# Motor imagery involves predicting the sensory consequences of the imagined movement

Konstantina Kilteni [1], Benjamin Jan Andersson[1], Christian Houborg[1] & H. Henrik Ehrsson[1]

Research on motor imagery has identified many similarities between imagined and executed actions at the behavioral, physiological and neural levels, thus supporting their "functional equivalence". In contrast, little is known about their possible "computational equivalence"—specifically, whether the brain's internal forward models predict the sensory consequences of imagined movements as they do for overt movements. Here, we address this question by assessing whether imagined self-generated touch produces an attenuation of real tactile sensations. Previous studies have shown that self-touch feels less intense compared with touch of external origin because the forward models predict the tactile feedback based on a copy of the motor command. Our results demonstrate that imagined self-touch is attenuated just as real self-touch is and that the imagery-induced attenuation follows the same spatiotemporal principles as does the attenuation elicited by overt movements. We conclude that motor imagery recruits the forward models to predict the sensory consequences of imagined movements.

---

[1] Department of Neuroscience, Karolinska Institutet, Retzius väg 8, 17177 Stockholm, Sweden. Correspondence and requests for materials should be addressed to K.K. (email: konstantina.kilteni@ki.se)

One of the most remarkable abilities of the human mind is its capacity to mentally simulate actions without physically executing them. More than two decades ago, Jeannerod and Decety[1,2] proposed that imagined movements are the internal simulation of actual movements. This influential "simulation hypothesis" received impressive support from experimental studies: imagined movements were shown to have similar durations[1,3–6] and to follow the same trade-off between movement duration and task difficulty as real movements[7–10]. Physiological studies revealed that imagined movements can increase physiological variables (e.g., heart rate) proportionally to the imagined effort, similar to executed movements[11,12]. Neuroimaging studies further showed that motor imagery activates a set of frontal motor areas, parietal areas, and cerebellar regions that partially overlaps with the brain network that is activated during motor execution[13,14] (for reviews, see refs. [15,16]), and that motor imagery of different effectors activates the corresponding sections of the somatotopically organized motor cortex[17]. This behavioral, physiological, and neuroimaging evidence enforced the view that imagined movements are functionally equivalent to the executed ones in terms of intentions, motor planning, and engagement of motor programs[14,18]. However, a strict prediction of the simulation hypothesis is that motor imagery should engage the same mechanisms in terms of predictive computational units ("forward models", see next paragraph) to generate sensorimotor predictions as real movements. To date, no previous study has directly tested this hypothesis of computational equivalence between motor imagery and motor execution; therefore, we do not know whether imagined movements engage the same central sensorimotor mechanisms as overt movements do. This question is fundamental not only for our basic understanding of the neurocognitive basis of motor imagery but also for emerging applications that seek to use motor imagery as a tool to control brain machine interfaces and advanced prosthetic limbs[19–21], or to replace or complement physical therapy and mobility training in the rehabilitation of neurological patients[22–24].

The prevailing theories of motor control posit that our ability to execute quick and accurate movements relies on computational units called forward models. These models encode the dynamics of our body parts in their interactions with the environment and anticipate the outcome of our voluntary movements[25,26]. When a motor command is sent from the motor cortices to the muscles, a copy of that motor command (efference copy) is also sent to the forward models. The forward models then use this information about the motor command to predict both the state of our body after the upcoming movement and the sensory consequences that the movement is likely to generate (sensory predictions)[27]. These predictions are effectively combined with the actual sensory feedback to provide a more reliable estimate of the state of the body compared with what can be derived from the sensory feedback alone, because such afferent information is both noisy and delayed[28]. In addition, the brain uses these predictions to attenuate the perception of the sensory feedback of the movement (reafference), thereby increasing the salience of stimuli that are generated by external causes[29,30]. The classical example of this phenomenon is that when we touch one hand with the other, the touch feels weaker compared with a touch of identical intensity that is applied by another person or a robot. This occurs because the self-generated touch has been predicted by the forward models based on the efference copy; consequently, the somatosensory feedback is attenuated[31,32]. The sensory attenuation of self-touch is thus a well-established paradigm for probing the engagement of efference copy and forward models in sensorimotor control.

To test the hypothesis that motor imagery engages the same computational mechanisms as real movements do, we here examined whether imagined self-touch elicits somatosensory attenuation. Specifically, we examined whether an external touch applied to the left index finger is attenuated when participants simultaneously imagine a voluntary movement of pressing their right index finger against their left one. Our experiments show that imagined self-touch is attenuated and that this imagery-driven attenuation has a comparable magnitude to the attenuation produced during overt movements. Further control experiments demonstrate that the imagined movement has to comply with the spatial and temporal principles of self-touch for sensory attenuation to be observed. We conclude that during motor imagery, the forward models predict the sensory consequences of the imagined movement.

## Results

**Force-matching task**. To psychophysically quantify somatosensory attenuation, we used the force-matching task[33]. In the classic version of this task, in each trial, participants receive a briefly presented force on the pulp of their relaxed left index finger (reference force) by a probe controlled by a DC motor. Immediately afterwards, they are asked to produce a force that matches the reference force in terms of magnitude (matched force). When asked to generate the matching force by pressing the pulp of their right index finger against a sensor placed directly above their left index finger—thus effectively simulating self-touch —participants consistently apply forces that are stronger than the reference ones[30,33–35]. This shows that the self-generated force feels weaker than the externally produced reference force, and that the participants are compensating for this by increasing the level of applied force to match the previously experienced reference force level. This effect thus constitutes objective evidence for somatosensory attenuation resulting from the sensory predictions generated by the forward models. In contrast, when the participants use a slider (or a joystick) that controls the force output on their left index finger, they are accurate in matching the reference forces[30,33–35]. In this baseline condition, the motor command (which is generated to horizontally displace the slider) and the force output are unusually related; therefore, the forward model cannot reliably predict the force sensations and no attenuation occurs.

To allow us to study the effects of motor imagery on somatosensory attenuation, we modified the force-matching task so the experimental manipulation occurred during the application of the reference force instead of during the generation of the matched forces (Fig. 1a–c). In the press condition, the participants pressed their right index finger against a sensor placed above (but not in contact with) their left index finger as strongly as they felt was required to match the reference force that they simultaneously felt on their left index finger (Fig. 1b). In the imagine condition, the participants were instructed to imagine that they were pressing their right index finger against the sensor and to imagine a force level that corresponded to the one they concurrently felt on their left index finger (Fig. 1c). Finally, in the base condition, the participants completely relaxed their right hand and right index finger without imagining any action (Fig. 1a). To report the perceived intensity of the reference forces applied to the left index finger, the participants used a slider with their right hand to generate the matched forces in all conditions (Fig. 1a–c). We hypothesized that if the forward models predicted the sensory consequences of the imagined finger pressing—as they do for executed pressing—then the reference force applied on the left index finger in the imagine condition would feel less intense compared with the identical forces exerted in the base condition.

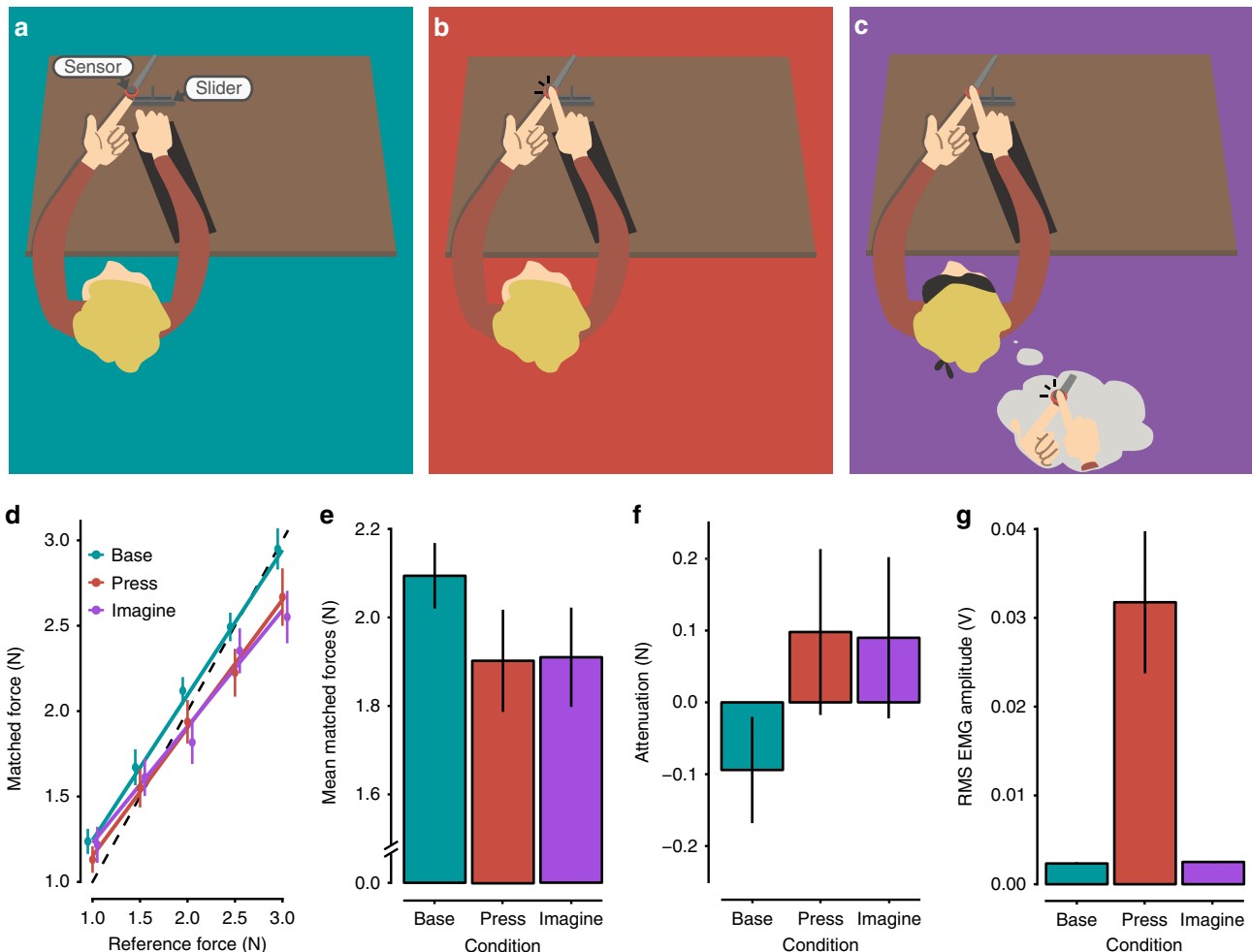

**Fig. 1** Conditions of Experiment 1 and Results. **a–c** Participants received a reference force on their relaxed left index finger by a probe attached to a lever controlled by a DC motor. During the application of this reference force (3 s), the participants were instructed to (i) keep their right hand and right index finger relaxed on top of a support (base; **a**); (ii) press a sensor with the right index finger (press; **b**); or (iii) imagine pressing the sensor with their right index finger (imagine; **c**). When pressing or imagining pressing, they were instructed to use as much force as they felt was required to match the reference force that they simultaneously felt on the left index finger. Immediately afterwards, they were asked to reproduce the reference force by using a slider that controlled the force output on their left index finger. **d** Forces generated by the participants (matched forces) as a function of the reference force. Points represent the matched forces of the participants by condition, averaged across the repetitions of each reference force level. Errors represent the SEM ( ± SE). The dotted line indicates the theoretically perfect performance. Colored lines represent the fitted regression lines per condition. For illustration purposes, the position of the markers has been horizontally adjusted to avoid overlapping points (for detailed statistical results see main text). **e** Matched forces per condition, averaged across the reference force levels. Error bars represent the SEM ( ± SE). Participants produced significantly weaker forces ($p < 0.05$) when matching the previously felt reference force on the left index finger during the press or imagine conditions compared to the baseline (base). **f** Somatosensory attenuation displayed per condition; here, the matched forces are subtracted from the reference forces (mean ± SE). **g** Root-mean-square EMG activity of the right FDI muscle during the application of the reference forces per condition, averaged across all trials. Errors represent the SEM ( ± SE). No significant difference was detected between the base and imagine conditions, demonstrating that the FDI muscle remained relaxed during the motor imagery

Twelve naive, right-handed, healthy volunteers participated in Experiment 1. Figure 1d shows the participants' performance in the three conditions across the different levels of reference force that were tested. A repeated-measures analysis of variance (ANOVA) revealed a significant main effect of condition ($F$(2,22) = 4.68, $p = 0.020$), a significant main effect of reference force level ($F$(4,44) = 153.5, $p < 0.001$), and a significant interaction ($F$(8,88) = 2.81, $p = 0.008$). Residual errors were normally distributed (Shapiro–Wilk test, $p = 0.297$). Pairwise comparisons between the levels of the reference forces revealed significant differences for each pair ($p < 0.001$), which confirmed that participants discriminated each reference force level well. As seen in Fig. 1e, the participants produced weaker forces both when they had previously pressed (mean ± SD = 1.902 ± 0.400 N)

or imagined pressing (mean ± SD = 1.910 ± 0.389 N) the sensor compared with the base condition, when they relaxed their index finger without imagining (mean ± SD = 2.094 ± 0.256 N). A pairwise comparison revealed that participants generated significantly weaker forces in the press condition compared with the base control condition ($t$(11) = –2.46, $p = 0.032$, confidence interval (CI) = [– 0.364, – 0.020], Hedges' $g_{av}$ = 0.565), confirming that when participants actively pressed the sensor on top of their left index finger, the reference force was attenuated and thus felt weaker. Critically, the imagine condition also yielded significantly weaker forces compared with the base condition ($t$(11) = – 2.76, $p = 0.018$, CI = [– 0.331, – 0.038], Hedges' $g_{av}$ = 0.551), which showed that the reference force felt on the left index finger was attenuated when participants simultaneously

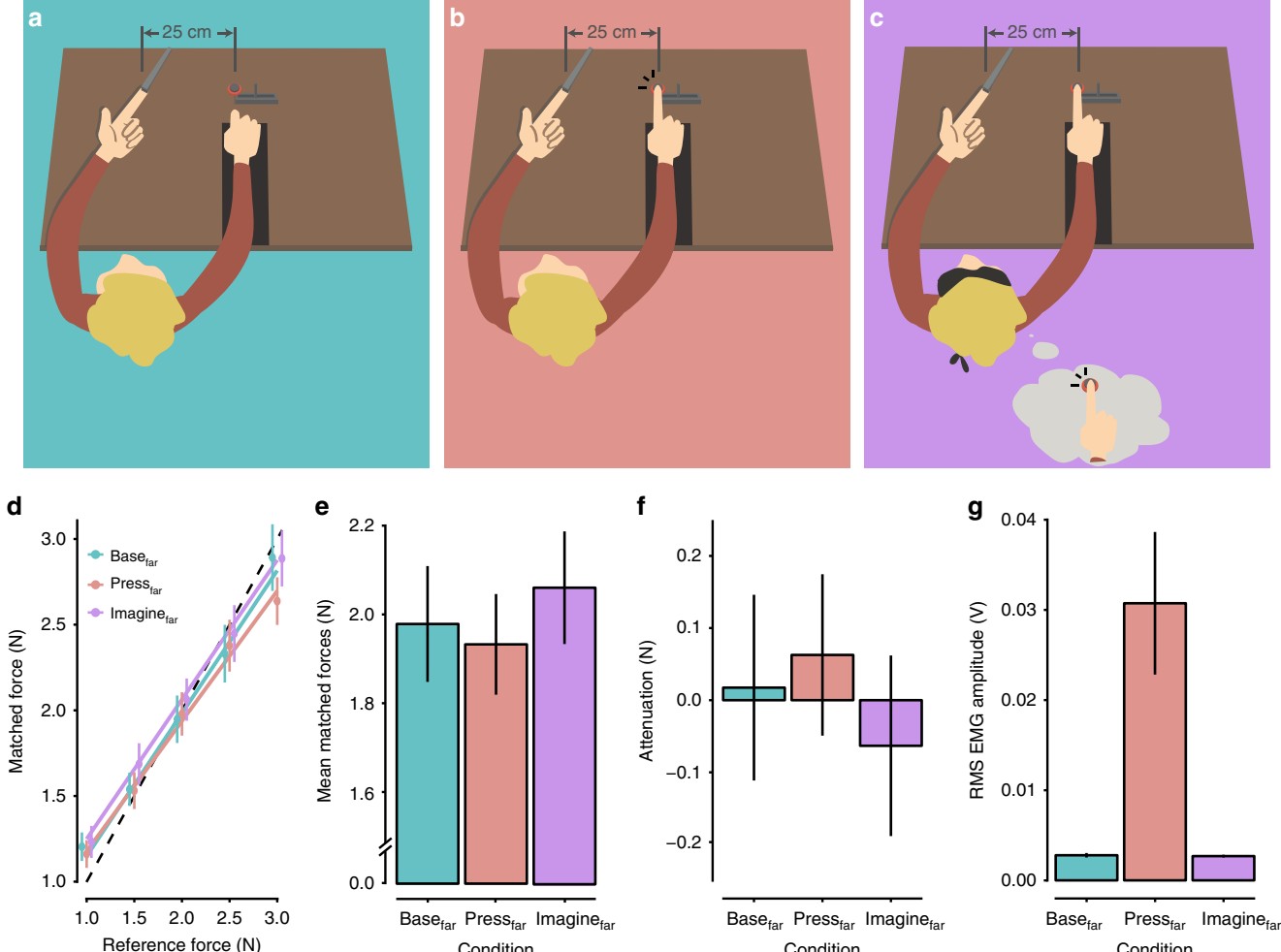

**Fig. 2** Conditions of Experiment 2 and Results. **a–c** The conditions of Experiment 2 were identical to those of Experiment 1, with the only difference being that the participants' right hands, the sensor, and the slider were all placed 25 cm to the right of their left index fingers. As it has been shown that introducing such a distance between the hands eliminates somatosensory attenuation, we tested whether the same holds true for motor imagery. **d** Forces generated by participants (matched forces) as a function of the reference force (mean ± SE). There was no significant effect of condition on the participants' performances, i.e., we found no evidence for somatosensory attenuation (see main text for detailed results). **e** Mean matched forces displayed per condition (mean ± SE). There were no significant differences among the three conditions. **f** Somatosensory attenuation expressed as the difference between the reference forces and the matched forces per condition (mean ± SE). **g** Root-mean-square EMG activity of the right FDI muscle averaged across all applications of the reference force per condition (mean ± SE). Importantly, no significant difference was detected between the base$_{far}$ and imagine$_{far}$ conditions, showing that the participants were able to relax their hand in the motor imagery condition

imagined pressing their right index finger against the sensor. Remarkably, the participants' forces did not differ significantly between the press and imagine conditions ($t(11) = -0.12$, $p = 0.908$, CI = [−0.157, 0.141], Hedges' $g_{av} = 0.019$), showing that the imagery-induced attenuation had a similar magnitude as the attenuation produced by real force production. Figure 1f illustrates these somatosensory attenuation effects, expressed as the difference between the reference forces and the matched forces per condition.

To ensure that the participants did not generate any small muscular contractions while imagining—a factor that would confound motor imagery with motor execution[36]—we also recorded surface electromyographic activity (EMG) from the first dorsal interosseous (FDI) muscle of the right hand during all conditions. Figure 1g shows the root-mean-square (RMS) of the EMG activity averaged across all applications of reference forces per condition. As expected, the FDI activity was greater during the press condition (mean ± SD = 0.032 ± 0.028 V) compared with the base (mean ± SD = 0.002 ± 0.0005 V) and imagine

conditions (mean ± SD = 0.003 ± 0.0004 V). Pairwise comparisons confirmed that the FDI was significantly more active during the press compared with both the base condition ($n = 12$, $V = 78$, $p < 0.001$, CI = [0.016, 0.036], $r = 0.624$) and the imagine condition ($n = 12$, $V = 78$, $p < 0.001$, CI = [0.016, 0.037], $r = 0.624$). Critically, there was no significant difference between the base and imagine conditions ($t(11) = -1.23$, $p = 0.243$, CI = [−0.0005, 0.0001], Hedges' $g_{av} = 0.36$). These results confirm that the participants were able to relax their index finger while imagining pressing it against the sensor.

In summary, Experiment 1 showed that the motor imagery of pressing the right index finger against the sensor above the left index finger produced somatosensory attenuation. To confirm that this imagery-induced attenuation was due to the predictions of the forward models and not to general factors related to performing mental imagery, such as differences in the attentional demands between the base and imagine conditions (e.g., divided attention), we conducted a control experiment. We hypothesized that if the sensory attenuation observed in the imagine condition

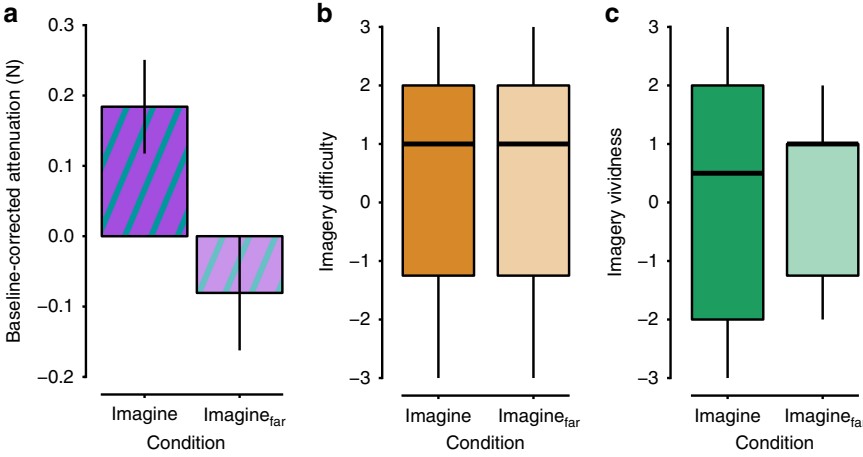

**Fig. 3** Comparing attenuation and task performance across the imagery conditions of the two experiments. **a** Baseline-corrected attenuation for the imagery conditions used in the two experiments (mean ± SE). There was significantly stronger attenuation ($p < 0.05$) during imagery of pressing the sensor when the sensor was placed on top of the left index finger compared with when it was placed at a distance of 25 cm from left index. **b**, **c** Boxplots for the ratings of difficulty and vividness of the motor imagery. After each experiment, we asked the participants to rate, on a 7-point Likert scale, how difficult they experienced the task and how vivid their mental imagery was. The horizontal black bars represent the medians. The boxes represent the interquartile ranges. The whiskers extend to the lowest and highest value within the 1.5 × interquartile range. There were no significant differences in imagery difficulty or imagery vividness between the groups, thus supporting similar performance

of Experiment 1 is truly a consequence of the predictions of the forward model based on the efference copy, then imagining another movement that does not have tactile consequences for the left hand should not produce attenuation. Therefore, in Experiment 2, a new group of 12 naive, right-handed volunteers participated under exactly the same conditions as those of Experiment 1 (identical instructions and task), with the only difference being that their right hand, the sensor, and the slider were all displaced by 25 cm to the right of their left index finger (Fig. 2a–c). Previous studies[30,34,37] have shown that when such a distance is introduced between the hands, touch is perceived more accurately, because the two hands are too far apart for the forward models to reliably anticipate physical contact and to predict touch[34]. Thus, we hypothesized that if imagery-induced attenuation is due to the sensory predictions of the forward models, then in this control experiment, we should observe no differences between the base and imagine conditions, because the forward models would no longer generate the somatosensory predictions required for the attenuation to occur.

Figure 2d shows the participants' performance for each level of reference force in the three conditions of Experiment 2. There was no significant main effect of condition ($F(2,22) = 1.30$, $p = 0.292$) and no significant interaction between condition and reference force level ($F(8,88) = 1.40$, $p = 0.209$). Only a significant main effect of reference force level was detected ($F(4,44) = 208.7$, $p < 0.001$), confirming again that the participants could discriminate the reference force levels well (for all pairwise comparisons, $p < 0.001$). Residual errors were normally distributed (Shapiro–Wilk test, $p = 0.810$). Participants produced similar forces in the three conditions (base$_{far}$: mean ± SD = 1.983 ± 0.446 N; press$_{far}$: mean ± SD = 1.937 ± 0.388 N; imagine$_{far}$: mean ± SD = 2.063 ± 0.434 N) (Fig. 2e–f). Pairwise comparisons further revealed that the matched forces in the press$_{far}$ condition did not differ from the base$_{far}$ condition ($t(11) = -0.49$, $p = 0.634$, CI = [− 0.249, 0.158], Hedges' $g_{av} = 0.106$), confirming the previous findings that when the hands are placed at a distance, the sense of force is not attenuated[34]. Importantly, the forces in the imagine$_{far}$ condition did not differ significantly from those in the base$_{far}$ condition ($t(11) = 0.99$, $p = 0.345$, CI = [− 0.099, 0.260], Hedges' $g_{av} = 0.175$), which suggests that

no attenuation occurred during the motor imagery. In accordance with this finding, the forces in the imagine$_{far}$ condition were not significantly different from those in the press$_{far}$ condition ($t(11) = 2.12$, $p = 0.057$, CI = [− 0.005, 0.256], Hedges' $g_{av} = 0.296$). Finally, the analysis of the EMG data from Experiment 2 confirmed that the participants were able to relax their right hand and right index finger in the motor imagery and baseline conditions equally well ($n = 12$, $V = 38$, $p = 0.970$, CI = [− 0.0002, 0.0004], $r = 0.016$) (Fig. 2g). As expected, the FDI was significantly more active during the press$_{far}$ compared with the base$_{far}$ ($n = 12$, $V = 78$, $p < 0.001$, CI = [0.011, 0.049], $r = 0.624$) and imagine$_{far}$ conditions ($n = 12$, $V = 78$, $p < 0.001$, CI = [0.011, 0.049], $r = 0.624$). Taken together, these observations refute any concern that differences related to performing imagery per se (including the somatosensory aspects of kinesthetic motor imagery) or differences in attention or cognitive effort between the imagery and baseline conditions could explain the findings in Experiment 1; in contrast, they suggest that adding a distance between the hands eliminates the imagery-induced attenuation.

To directly test the effect of the spatial distance between the hands on the imagery-induced somatosensory attenuation, we directly compared the force data from the imagine conditions across the two experiments after normalizing the data to the corresponding base conditions (to eliminate unspecific between-group differences; Fig. 3a). The motor imagery conditions in the two experiments were matched in terms of rated vividness of the imagery ($n = 12$, $W = 76$, $p = 0.835$, $r = 0.048$) or difficulty in evoking the mental images ($n = 12$, $W = 71$, $p = 0.976$, $r = 0.012$) (Fig. 3b, c). Importantly, we found that the motor imagery of pressing the sensor when it was placed on top of the left index finger led to significantly greater attenuation compared with imagery of pressing the sensor when it was positioned at a 25-cm distance from the left index finger: $t(22) = 2.51$, $p = 0.020$, CI = [0.046, 0.484], Cohen's $d_s = 1.024$. This shows that the imagery-induced force attenuation follows the same spatial rule as does the attenuation generated by overt voluntary action.

Finally, we conducted an additional control experiment (Experiment 3) to address the temporal rule of the imagery-induced attenuation effect under investigation. Previous findings

**Fig. 4** Computational equivalence between imagined and executed movements. **a** Somatosensory attenuation associated with pressing one index finger against the other. Given a copy of the motor command, the forward models predict the next state of the body and the associated sensory consequences of this state. When pressing one finger against the other, these sensory consequences include the tactile feedback from the self-touch. The predicted touch is used to attenuate the actual tactile feedback (comparator). This model is based on an earlier proposal[38]. **b** Somatosensory attenuation during imagining pressing one index finger against the other. For the covert action, the forward models predict tactile feedback based on the efference copy that is generated as part of the internal simulation of the action. When touch is applied externally on the finger in a way that matches the predicted feedback from the imagined movement, the somatosensory sensation is attenuated, just as occurs during real self-touch, because the forward models have already predicted it

on the force-matching task have shown that somatosensory attenuation is observed when the forces applied to the receiving finger occur at the same time that the participants actively produce the forces on the sensor. If a large temporal delay is introduced between the two events, the attenuation effect is eliminated[31]. To determine whether the same principle holds true for motor imagery, we conducted an additional control experiment with twelve new volunteers (Supplementary Note 1). The results showed that the imagery-driven attenuation requires participants to perform the imagery task at the same time that they receive the reference force (Supplementary Note 1). No significant attenuation of the perceived forces was observed when the imagery task was performed five seconds before the application of the reference force (Supplementary Fig. 1); however, concurrent imagery and reference force replicated the results from Experiment 1. These findings further corroborate our main conclusion that the imagery-driven sensory attenuation is the result of predictions made by the forward model that require a likely causal relationship between the (imagined) movement and its sensory consequences according to the spatiotemporal constraints of self-touch perception[31,38].

## Discussion

The main conclusion of the present study is that motor imagery produces somatosensory attenuation just as real movements do. Specifically, our results showed that imagined finger pressing produced an attenuation of self-touch and that this imagery-induced attenuation was of comparable magnitude and obeyed the same spatiotemporal rules between the hands as overt force production. These findings not only support the notion of functional equivalence between motor imagery and motor execution, but they demonstrate a "computational equivalence". It has long been theorized that forward models are utilized during motor imagery[25,28,39]. Blakemore and Sirigu[40] postulated that during motor imagery, a forward model specific for the imagined action needs to be retrieved. Almost at the same time, Grush[41] introduced his 'emulation hypothesis,' according to which forward models (emulators) actively simulate the imagined action and predict its sensory consequences. Nevertheless, these proposals remained a conjecture[42,43], because the covert nature of imagery and its lack of sensory feedback have made this assumption difficult to experimentally test. Although a few experimental studies have suggested the involvement of predictive processes during speech imagery[44–47], there has not been

experimental evidence directly supporting or falsifying this theory. Given that the somatosensory attenuation emerges when the forward models predict the touch based on the efference copy[27,30,31,34,48], our findings bring direct and conclusive experimental behavioral evidence that motor imagery does indeed recruit the forward models to predict the sensory consequences of the imagined action. This constitutes a major advance in our understanding of the computational principles of motor imagery.

Based on our results, we propose that imagining a movement engages the same forward models as does physically executing the imagined action (Fig. 4a, b). Such a common mechanism would be computationally less expensive than having different mechanisms for overt and covert movements. When we imagine moving one hand to touch the other, the forward models simulate the imagined action based on the efference copy and predict the end states of the limbs after the imagined action as well as their sensory consequences, just as they do for real movements. If the predicted positions of the limbs after the imagined movement indicate contact between two body parts, then the forward models generate tactile predictions. Thus, in our experimental setup, when we delivered real tactile stimulation at the right time and at the right place on the body to coincide with the "effects" of the imagined finger action, the brain processed these somatosensory signals as though they were the actual anticipated sensory feedback of the imagined movement. Consequently, these somatosensory signals got attenuated, because they had been predicted by the forward models (Fig. 4b), which demonstrates that motor imagery entails the prediction of the sensory consequences of the imagined action.

What could be the neuronal mechanisms that underlie the current imagery-induced somatosensory attenuation effects? Previous studies have shown that motor imagery consistently activates regions related to motor execution such as the supplementary motor area, the premotor cortex and the cerebellum[16,49–57]. The motor programs and efference copy are likely produced in the non-primary motor areas and sent to the cerebellum via the anatomical connections that exist between these structures[58–60]. The cerebellum is then a strong candidate for the neural substrate of the forward models[61–64] that could generate internal sensory predictions for both executed and imagined actions.

The present results are also interesting from the perspective of computational sensorimotor control, because they provide unique experimental evidence supporting the critical role had by the efference copy in sensory attenuation. Although previous

computational models have postulated a central role for efference copy as an input signal to the forward models, e.g.,[27,28,30,38,65], conclusive experimental evidence in favor of this idea has been lacking. Efference copy is typically studied in conditions with active movement, but active movement additionally involves the descent of motor commands to the spinal cord, muscular contractions and proprioceptive feedback, as well as other kinds of somatosensory feedback (e.g., skin stretching, joint pressures). In previous experimental studies on the sensory attenuation of self-touch[30,31,33,34,37], the possible effect of efference copy was always confounded by these additional factors. This is problematic because somatosensory feedback during active overt movements could be subject to "sensory filtering" or "gating" mechanisms[66], which could reduce the perceived intensity of the somatosensory percepts, independent of the efference copy. Therefore, the present results are important because they suggest that the efference copy is sufficient to elicit the sensory attenuation of self-touch in the absence of any actual movement.

Finally, the present findings have far-reaching implications for theories of motor learning through mental rehearsal. There is a growing interest among movement researchers and clinical professionals to use motor imagery as a tool to enhance motor performance and facilitate motor learning. Several studies have shown that motor imagery can significantly improve motor performance in terms of, e.g., movement accuracy and efficacy (see refs. [67,68] for reviews). Moreover, it has been shown that this imagery-driven motor learning can generalize to similar tasks that are physically executed afterwards[69]. Indeed, motor imagery is one of the most common strategies for practicing among elite athletes[42,70] and professional musicians[71], and it has been proved to be beneficial for the motor rehabilitation of neurological patients[22–24]. Dominant theories of motor learning suggest that our brain uses the discrepancy between the feedback predicted by the forward models and the actual sensory feedback (sensory prediction error) to update the forward models and improve our control policies[72,73]. The present study indicates that the motor improvement seen after the mental practice of movements might be due to the forward models running offline based on the efference copy alone during imagery. During the repeated mental simulation of a movement, the forward models could use the difference between the predicted and the desired outcome as the teaching signal in the updating process, thereby establishing a better and finer motor performance on subsequent overt execution. Motor imagery thus truly corresponds to the mind's internal simulation of action that can be used to improve further performance and induce neural plasticity.

## Methods

**Participants**. Twelve healthy participants (5 women and 7 men, all right-handed) aged 22–37 years participated in Experiment 1, and 12 different healthy participants (6 women and 6 men, all right-handed) aged 18–30 years took part in Experiment 2. The sample size was chosen based on previous studies[31,33]. All participants were naive to the purpose of the studies. Three additional participants were recruited but excluded and replaced, because two of them (one from Experiment 1 and one from Experiment 2) were unable to fully relax their right hand and index finger in imagery and baseline conditions, as evident in the on-line EMG recordings; the third one was excluded after Experiment 1, because he reported that he did not perform the imagery task as instructed.

None of the participants reported a current or previous history of psychiatric or neurological conditions, and none of them had a history or current use of any psychoactive medication. The Regional Ethical Review Board of Stockholm approved both experiments, and all participants gave their written informed consent. Handedness was assessed using the Edinburgh Handedness Inventory[74].

**General procedure**. Participants rested their left hand palm up with their left index finger placed on a molded support. In each trial, a cylindrical probe (25 mm height) with a flat aluminum surface (20 mm diameter) attached to a lever that was controlled by a DC motor (Maxon EC Motor EC 90 flat, manufactured in Switzerland) applied a force on the pulp of the participants' left index finger. A small

force sensor (FSG15N1A, Honeywell Inc.; diameter, 5 mm; minimum resolution, 0.01 N; response time, 1 ms; measurement range, 0–15 N) was placed inside the probe. This sensor measured the reference forces and the matched forces applied on the participants' finger throughout the experiments. An identical force sensor (sensor displayed in Figs. 1 and 2) was placed inside a second cylindrical capsule that was positioned on top of a small wooden structure. The wooden structure with the capsule and the sensor was placed either on top of (but not in contact with) the probe of the lever above the left index finger (Experiment 1) or 25 cm to the right of the participants' left index fingers (Experiment 2).

The participants' right forearms and elbows were comfortably placed on top of two box-shaped cuboids made of sponge (length × width × height, 20 cm × 10 cm × 10 cm). The position and height of the boxes were carefully selected such that a part of the participants' hands could protrude from the boxes and their right index fingers could comfortably rest on top of the capsule without any muscular contraction and without touching anything else.

During each trial, the probe exerted a constant reference force on the participants' left index fingers (1 N, 1.5 N, 2 N, 2.5 N or 3 N) that lasted 3 s. At the same time, in the press condition, the participants pressed their right index finger against the sensor with force required to match the reference force that they simultaneously felt on their left index finger. The reference forces applied to the left index finger were unaffected by the forces simultaneously applied to the sensor by the participants' right index fingers, and the participants were explicitly informed about this so that it would make the task easier. Immediately after the presentation of each reference force trial, the participants moved the slider of a 13 cm linear slide potentiometer with their right hands. The slider was placed just in front of their right hands, so it was easy to reach and use. The slider controlled the force output of the lever on their left index fingers. The lower limit (left end) of the slider corresponded to 0 N and the upper limit (right end) corresponded to 5 N. Participants had 3 s to produce a force that matched the previously applied reference force. Upon completion of this period, participants were asked to return the slider to 0 N; thus, every trial started with the slider being at 0 N (left end). Participants were encouraged to adjust their response in the beginning of the period to find the force level that best matched the reference force but to keep their responses stable during the last second of the trials.

In both experiments, participants wore headphones through which moderately loud white noise was played to eliminate any possibility that the participants could hear the motor generating the forces. The onset and offset of the periods for the reference and matched forces were indicated with auditory "go" and "stop" signals. Moreover, during the base and press conditions of both experiments, participants were asked to fixate a black cross that was placed on the wall opposite them (2 m distance), whereas in the motor imagery conditions, they were blindfolded with a common sleep mask.

Each of the three experimental conditions (see main text above) consisted of 35 force trials, and each of the 5 reference force levels was pseudorandomly repeated 7 times. The order of the conditions was counterbalanced across participants. No feedback was ever provided to the participants concerning their performance during the experiments.

**Procedures in the imagery condition**. In the imagery conditions, the participants were instructed to imagine pressing their right index finger against the sensor, just as they actually pressed the finger in the press condition. We emphasized to the participants that they should imagine performing the movement from a first person perspective (and not from a third-person perspective) and should imagine the sensations associated with the movement (i.e., the feeling of force and muscle contractions in the finger (kinesthetic-motor imagery)) but should not visualize the movement as though they were watching a video[42,75] (visuo-motor imagery). To make it easier for the participants to understand exactly what action they were supposed to imagine and thereby increase the vividness of the motor imagery, all participants first practiced five to ten trials of the actual pressing condition before they practiced the corresponding motor imagery task. No feedback was ever provided to the participants concerning their performance during these training periods. Finally, before the actual force experiment commenced, we asked the participants whether they could reliably perform the requested motor imagery. All participants stated that they could perform the imagery task as instructed.

**Force registration, processing, and statistical inference**. For each trial, we calculated the average matched force participants generated with the slider during the period 2000–2500 ms after the "go" signal in accordance with previous studies[30,33]. During this time period, the applied force level had stabilized. We then averaged the forces across the seven repetitions of each level of reference force. Then, for each experiment, we conducted a repeated-measures ANOVA with factors (i) the level of the reference force and (ii) the condition. The normality of residual errors was assessed with the Shapiro–Wilk test. Planned pairwise comparisons were conducted with two-tailed paired $t$-tests, as the distributions confirmed normality. Finally, to compare the effects of imagery between the experiments, we first subtracted the matched forces of the imagery conditions from their corresponding base conditions and we then contrasted them with a two-tailed unpaired $t$-test, because the distributions were normal and had similar variances. Force data were analyzed using R (R version 3.3.2, RStudio Version 1.0.136).

**EMG collection, processing, and statistical inference**. Surface electromyography was recorded using the Delsys Bagnoli electromyography system (DE-2.1 Single Differential Electrodes) from the right FDI muscle after cleaning the skin with alcohol. The electrode was carefully placed over the belly of the muscle. The signals were analog bandpass-filtered between 20 and 450 Hz, sampled at 2.0 kHz and amplified (gain = 1000). An additional notch filter was used to suppress the 50 Hz power line interference and the DC offsets of the signals were removed.

For each trial, we calculated the RMS of the EMG signal during the full window (3 s) of the reference force. Then, we averaged the values across the 35 trials of each condition and for each participant. Planned pairwise comparisons were performed with either a two-tailed paired $t$-test or a two-tailed Wilcoxon signed-rank test, depending on the normality of the distributions. EMG data were processed in Matlab R2015 and analyzed using R (R version 3.3.2, RStudio Version 1.0.136).

In the base and imagine conditions of each experiment, participants were reminded to relax their hands and index fingers during the application of the reference force. To ensure that participants did not contract the muscles of the right index finger, the EMG signals were always displayed on a computer screen on-line and monitored throughout the experiment by a second experimenter. Trials in which there was visible muscular activity were rejected and repeated after reminding the participants to further relax their right hands and index fingers.

**Post-experiment questionnaires and statistical inference**. After each experiment, participants were asked to rate their level of agreement for two statements on a 7-point Likert scale, ranging from − 3 (strongly disagree) to + 3 (strongly agree), with 0 indicating "neither agree nor disagree". The two statements were as follows:

Statement 1 (S1). I found it difficult to imagine moving my finger during the experiment.

Statement 2 (S2). I found that when I imagined moving my finger, the movement seemed clear and vivid, almost as if the movement were real.

To test for differences in the questionnaire responses between the two experiments, we used a two-tailed Mann–Whitney $U$-test.

Effect sizes were estimated with Hedges $g_{av}$ for paired $t$-tests and with Cohen's $d_s$ for unpaired $t$-tests, as suggested by ref. [76]. For the Mann–Whitney $U$-test and Wilcoxon's signed-rank test, effect sizes were calculated as $r = Z/\sqrt{N}$, where $N$ is the total sample size of the given test (i.e., $N = 24$).

**Data availability**. The data that support the findings of this study are available from the corresponding author upon reasonable request.

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

## Acknowledgements

Konstantina Kilteni was supported by the Marie Skłodowska-Curie Intra-European Individual Fellowship (704438). The project was funded by the Swedish Research Council, Torsten Söderbergs Stiftelse, and Riksbanken Jubileumsfond.

## Author contributions

K.K. and H.H.E. conceived and designed the experiments. K.K. and B.J.A. collected the data from Experiment 1 and Experiment 2 together. K.K. and C.H. collected the data from the Experiment 3 (Supplementary Information) together. K.K. conducted the statistical analysis. K.K. and H.H.E. wrote the manuscript. B.J.A. and C.H. read and approved the final version.

## Additional information

**Competing interests:** The authors declare no competing interests.

