## [Peer Review File · Nature Communications]

Reviewers' comments:

Reviewer #1 (Remarks to the Author):

This study investigated the internal forward model in motor imagery by asking whether imagined self-action can induce somatosensory suppression. I believe it is an interesting and important question. However, fruitful research with similar goals have been carried out in recent years in relevant and more complex motor control context, especially in the field of speech production. Therefore, this study can be considered as a branch with a focused scope in the whole research domain of internal forward model. Moreover, using imagery to investigate internal forward model is a fairly innovated (only a few studies available) but a hard paradigm, which requires better controls. This study implemented a concurrent imagery-stimulation paradigm in which the dynamics and timing of imagery is hard to control. More careful and improved experimental design may reduce the complication of the results. I listed my detail comments below.

First, this study is motivated by classical motor imagery research mostly from Decety and Jeannerod groups, in 80s, 90s and early 2000s, and Wolpert and Blakemore studies about internal forward model and somatosensory suppression also in 90s and 2000s. Other references cited are mostly in 2000s too. These studies cannot reflect recent fast advance in this field. Moreover, as the title and introduction indicate that the authors would like to make a rather general claim about sensory consequence prediction in motor imagery, only including motor imagery in somatosensory domain would be too narrow to extend to a broader and significant research field of internal forward model and sensory consequence prediction in motor imagery. In fact, this topic has been explored in details in some relative research domains, for example in speech production, as well as in different sensory modalities. For example, it can be evident from their proposed model in Fig. 4, where the plot a (action part) was adapted from Blakemore, Frith and Wolpert, 2001 Fig.1, and the plot b is the same model as in Tian and Poeppel, 2010 and 2012 Fig. 1 and 3.

One of the research topics with similar goals is internal forward model and motor imagery in speech. Using imagined speaking paradigms, researchers have been investigating how the internal forward model predicts the sensory consequences – the same as the present study but in an even more complicated imagery and more direct motor control context. It has been demonstrated that imagined movement of articulators can predict consequences both in somatosensory and auditory domains. Electrophysiological and neuroimaging studies have shown that imagined speaking can activate somatosensory and auditory cortices in a successive temporal manner [Tian & Poeppel, 2010; Tian, Zarate & Poeppel, 2016], and can modulate auditory responses depending on the similarity between the imagined speech and the external stimuli [e.g. Tian & Poeppel, 2013; 2015; Herholz, Halpern & Zatorre, 2012]. Behaviorally, imagined speech can also influence speech perception [e.g. Scott, Yeung, Gick & Werker, 2013; Scott, 2013; 2016]. This is consistent with the finding about sensory prediction and internal forward model using overt speech paradigms, including speech-induced suppression (SIS, e.g. research from John Houde's group) and feedback perturbation and compensation (e.g. research from Charles Larson's group).

Moreover, the effects of imagery on perception has been also demonstrated in many sensory modalities, including in vision at all levels from high level spatial configurations (Tartaglia et al., 2009, *Curr Bio*), to lower level attributes such as orientation (Pearson et al., 2008, *Curr Bio*; Pearson et al., 2011, *Psych Sci*) - and even muscle control and pupil contraction (Laeng and Sulutvedt, 2014, *Psych Sci*), in auditory domain of syllable-level representation (Scott, 2013, *Psych Sci*) or pitch (Borra et al, 2013, *PNAS*), and in olfactory [Djordjevic, Zatorre, Petrides, and Jones-Gotman, 2004; Djordjevic, Zatorre and Jones-Gotman, 2004; Zelano, Mohanty and Gottfried, 2011], and even in food consumption [Morewedge, Eun Huh, Vosgerau, 2010].

The authors should at least include some of the above-mentioned recent studies that target at the similar research question. It can greatly increase the validity as well as the significance of this study.

Second, imagery can provide a useful paradigm to investigate mental representation. However, because the nature of imagery which is covert and without any external cues, it also introduces a lot of problems that need to control. For example, how can the authors be sure that participants performed the motor imagery as instructed? More importantly, this study implemented a concurrent imagery-stimulation paradigm. How can the timing and dynamics of motor imagery consistent with the occurrence of external stimulation? This is important because it has been demonstrated that the prediction generated from internal forward model is very precise. In order to observe the prediction-induced suppression effect, the prediction should be well aligned with external stimulation both in terms of timing and dynamics, as well as frequency and magnitude [e.g. Houde, Nagarajan, Kekihara & Merzenich, 2002; Tian & Poeppel, 2015]. If there is any deviance between prediction and external stimulation, enhancement effects could be observed instead [e.g. Behroozman, Karvelis, Liu & Larson, 2009; Tian & Poeppel, 2015]. In the current study, an external constant force was applied to participants' left index finger, but the self-press or imagined press arguably cannot remain on a consistent level or start/end at the same time as the external force (the author can check their pressing data which is not provided). Therefore, there were very likely deviance between the prediction and external stimulation, and hence possibly created complicated suppression as well as enhancement. The authors may need to perform some follow-up studies that either have better control of performance timing and dynamics, or have a better procedure that bypass these difficulties. One suggestion is to use repetition/adaptation paradigm in combination with mental imagery (e.g. Tian & Poeppel, 2013).

There is one particular setting of experiments that may seriously undermined the validity of this study. In this current study, the probe to the left index finger is independent from the force exerted from the right hand, and participants were aware that the two forces were unrelated (Pg. 11). This setting violates the critical assumption about internal forward model of sensory prediction, which is that the action is the cause of the sensory consequence. This unrelated setting between the button-press and external force makes the paradigm and results about internal forward model of sensory prediction in serious doubts.

Minor points:

I would not call the aim of this study 'computational equivalence' as compared to 'functional equivalence', because the observed imagery-induced perceptual suppression is the function of internal forward model – predicting the perceptual consequence of actions.

Why did the base condition show negative attenuation in experiment 1 but no attenuation in experiment 2? Should they be the same since they are rest conditions only differ in the position of rest?

Why did the authors choose a between subject design? Though the authors used base conditions to normalize the attenuation effect in imagery condition, it can still introduce some confounds. For example, it may be that base and imagery have complicated interaction caused by the distance of button placed. Therefore, a within subject design would be better.

Reviewer #2 (Remarks to the Author):

In the present manuscript, the authors show by means of two behavioral experiments that motor imagery of pressing the right index finger against a sensor produces sensory attenuation, as does actually pressing the right finger against the sensor. They argue that the imagery-induced attenuation is due to sensory predictions of the forward models of the imagined motor action, an interpretation based on contemporary theories on forward models in the field of action execution. The manuscript is extremely well written, and its conclusions are novel. I have however some important statistical/theoretical remarks that may prevent publication, at least in its current state.

One important remark concerns the fact that the authors' main conclusions are based on the absence of a statistical difference between two conditions, namely the 'press' and 'imagine' conditions. I think this poses problems for two reasons: For one, the authors bump into an interpretation problem that is reminiscent of the 'reverse inference' problem in the fMRI literature, which is in my opinion problematic. Based on the presence of similar behavioral/neural effects in two conditions, one cannot argue that identical cognitive mechanisms are necessarily underlying the effects. It might indeed be the case that a similar cognitive mechanism is underlying the attenuation of touch within both conditions, but this is not necessarily the case, logically speaking. It might also be the case that very distinct cognitive mechanisms produce a similar behavioral effect. How would the authors respond to this? Second, the authors do not provide measures that directly test and quantify the absence of a statistical difference (e.g., Bayesian tests or equivalence test). While the sample size of the experiments is relatively small (12 individuals in each experiment), subtle differences between the two conditions (that might have been present when a larger sample would have been used) might not have been detected in the current experiments.

Another major remark is related to the problem of specificity of the presumed efference copy. If the sensory attenuation that was observed in the imagine conditions is truly consequent to efference copies, another imagined movement that has different tactile consequences should not produce touch attenuation. I wonder, in other words, whether the attenuation is due to the fact that participants imagine the specific movement they are asked to imagine (i.e. pressing the right index finger against the sensor), or to the fact that they imagine touch per se (which could have been produced by any movement). While the authors would like to argue that the attenuation in the imagine condition is consequent to efference copies, this is an important question to ask. To solve this issue, I am afraid an additional control experiment will be needed. I suggest to manipulate the congruency of the imagined action (e.g. index finger or middle finger movement) and the felt touch (i.e., felt touch on index finger or middle finger; for a similar discussion on movement specificity in observed finger movements and felt touch, see Deschrijver, Wiersema & Brass, 2015; 2016). Only in the case where the sensory attenuation would be observed in congruent trials but not in incongruent trials, the authors would be able to effectively argue that tactile consequences of the specific imagined movement may underlie the attenuation effects.

On a similar note, one could wonder to which extent participants imagined the touch itself that followed the imagined movement, next to the movement alone. While the authors did not provide a post-Likert measure for the vividness of touch imagery (e.g., "I could 'feel' how the imagined movement pressed my finger." Or "I found that when I imagined moving my finger, the touch seemed clear and vivid, almost as if the touch were real"), the likelihood of this potential alternative explanation to interpret the data is difficult to assess. However, if true, one would not need to infer an explanation in terms of efference copies: the extent to which the participant imagined the touch consequent to the action would explain the observed effects. For this reason, I would suggest to include a post measure of imagined vividness of touch imagery in the control experiment: if this measure is found unrelated to the attenuation effects, an interpretation in terms of efference copies

may be more likely (as an efference copy should theoretically be present anyway following the imagined movement).

Additionally, I have some more minor comments:

- What are the effect sizes of the findings? I suggest to add these to the manuscript.
- It is not entirely clear to the reader what is meant exactly with terms like 'computational mechanisms' or 'predictive computational units'. I suggest to elaborate on this.
- The authors use their model to make theoretical claims about the role of efference copy for imagined movements. However, it should be noted that the findings are based on movements that involve a clear tactile consequence (i.e. a pressing movement). I think the theoretical implications should therefore be limited to efference copies of these specific types of movements, rather than to efference copies of imagined movement per se.
- It is unclear to me why the authors blindfolded participants in the imagine conditions but not in the baseline and movement conditions. This leads to a difference between conditions in the sense that that behavioral measures in the imagine condition are based on unimodal information (tactile), whereas those of the movement/baseline condition are based on multimodal information (visuotactile). Though I do understand that even in these multimodal conditions, participants did not visually perceive their hands, this does account for a modality difference over conditions in the methodological design. In addition, the unimodal findings (based on the tactile sense only) in the imagine condition are thus compared to a multimodal baseline condition (which was also visuotactile). Could the authors comment on why this methodological choice was made? While it is not entirely clear to me to which extent this choice could have influenced or biased the results, I think this should be noted to the attentive reader (and be avoided in future experiments)

Reviewer #3 (Remarks to the Author):

In this paper the authors demonstrate that imagined self-touch is attenuated just as real self-touch is and that the imagery-induced attenuation is of the same magnitude and follows the same spatial principle as does the attenuation elicited by an overt movement.

This paper is near flawless. The question is important, the methods are sound, the results are unambiguous, the control study nicely adds to the main findings, which are novel and impactful, and the discussion is very insightful. I only have minor comments to enhance the manuscript.

Minor comments

This sentence in the intro is a bit contradictory with respect to M1, as it leaves the reader unclear as to whether M1 activity is expected or not during imagery.

"Neuroimaging studies further showed that motor imagery activates a set of non-primary motor areas, parietal areas, and cerebellar regions that partially overlaps with the brain network that is activated during motor execution^{9,25} (for reviews, see^{6,7}) and that motor imagery of different effectors activates the corresponding sections of the somatotopically organized motor cortex⁵."

Line 179: "reference forces"

Line 195: "in exactly the same conditions"

Figure 3C: can't see the median...

Pierre-Michel Bernier

Reviewers' comments:**Reviewer #1 (Remarks to the Author):**

(1) First, this study is motivated by classical motor imagery research mostly from Decety and Jeannerod groups, in 80s, 90s and early 2000s, and Wolpert and Blakemore studies about internal forward model and somatosensory suppression also in 90s and 2000s. Other references cited are mostly in 2000s too. These studies cannot reflect recent fast advance in this field. Moreover, as the title and introduction indicate that the authors would like to make a rather general claim about sensory consequence prediction in motor imagery, only including motor imagery in somatosensory domain would be too narrow to extend to a broader and significant research field of internal forward model and sensory consequence prediction in motor imagery. In fact, this topic has been explored in details in some relative research domains, for example in speech production, as well as in different sensory modalities. For example, it can evident from their proposed model in Fig. 4, where the plot a (action part) was adapted from Blakemore, Frith and Wolpert, 2001 Fig.1, and the plot b is the same model as in Tian and Poeppel, 2010 and 2012 Fig. 1 and 3.

One of the research topics with similar goals is internal forward model and motor imagery in speech. Using imagined speaking paradigms, researchers have been investigating how the internal forward model predicts the sensory consequences – the same as the present study but in an even more complicated imagery and more direct motor control context. It has been demonstrated that imagined movement of articulators can predict consequences both in somatosensory and auditory domains. Electrophysiological and neuroimaging studies have shown that imagined speaking can activate somatosensory and auditory cortices in a successive temporal manner [Tian & Poeppel, 2010; Tian, Zarate & Poeppel, 2016], and can modulate auditory responses depending on the similarity between the imagined speech and the external stimuli [e.g. Tian & Poeppel, 2013; 2015; Herholz, Halpern & Zatorre, 2012]. Behaviorally, imagined speech can also influence speech perception [e.g. Scott, Yeung, Gick & Werker, 2013; Scott, 2013; 2016]. This is consistent with the finding about sensory prediction and internal forward model using overt speech paradigms, including speech-induced suppression (SIS, e.g. research from John Houde's group) and feedback perturbation and compensation (e.g. research from Charles Larson's group).

Moreover, the effects of imagery on perception has been also demonstrated in many sensory modalities, including in vision at all levels from high level spatial configurations (Tartaglia et al., 2009, Curr Bio), to lower level attributes such as orientation (Pearson et al., 2008, Curr Bio; Pearson et al., 2011, Psych Sci) - and even muscle control and pupil contraction (Laeng and Sulutvedt, 2014, Psych Sci), in auditory domain of syllable-level representation (Scott, 2013, Psych Sci) or pitch (Borra et al, 2013, PNAS), and in olfactory [Djordjevic, Zatorre, Petrides, and Jones-Gotman, 2004; Djordjevic, Zatorre and Jones-Gotman, 2004; Zelano, Mohanty and Gottfried, 2011], and even in food consumption [Morewedge, Eun

Huh, Vosgerau, 2010].

The authors should at least include some of the above-mentioned recent studies that target at the similar research question. It can greatly increase the validity as well as the significance of this study.

We thank the reviewer for his/her positive comments, and we are very grateful for the suggested literature.

We acknowledge that there has been previous work in the speech domain supporting the existence of prediction during imagery. Particularly relevant for current purposes is the neurophysiological¹ and neuroimaging² work of Tian, Poeppel and colleagues, as well as the behavioral work by Scott³ that showed that speech imagery affects the influence of a concurrently heard syllable on a subsequently heard syllable, arguing that this effect occurs because the first syllable was attenuated. We now cite these studies in our manuscript as studies that support the involvement of forward models during imagery (Lines 300-302).

We also acknowledge that there is a plethora of studies (including most of the suggested ones but many more) that have systematically shown the effects of motor and sensory imagery on perception, e.g., on perceptual learning, facilitation of conscious perception, unisensory or multisensory perception, etc. Our group has also previously shown effects of visual and auditory imagery on unimodal and multimodal perception^{4,5}. However, we would like to emphasize that (a) our study is specific to kinesthetic motor imagery and (b) its purpose is very specific to testing whether the forward models predict the sensory consequences of the imagined movement. These points are why our citation list must be limited to this specific question subarea.

We definitely agree with the reviewer that the involvement of forward models in motor imagery has been theorized in previous theoretical papers (as reflected also in our text at Lines 294-299), but it remains a hypothesis in the sense that there is no *direct* experimental evidence for this research question. We designed the present study to precisely provide such behavioral evidence, and we consider that our findings clearly support the model. We observed a very specific pattern of behavioral results that precisely follows the predictive computations of the internal models, as opposed to the mere general behavioral similarities between motor imagery and motor execution that have been described previously.

References

1. Tian, X. & Poeppel, D. Mental imagery of speech and movement implicates the dynamics of internal forward models. *Front. Psychol.* (2010). doi:10.3389/fpsyg.2010.00166
2. Tian, X., Zarate, J. M. & Poeppel, D. Mental imagery of speech implicates two mechanisms of perceptual reactivation. *Cortex* **77**, 1–12 (2016).
3. Scott, M. Corollary Discharge Provides the Sensory Content of Inner Speech. *Psychol. Sci.* **24**, 1824–1830 (2013).
4. Berger, C. C. & Ehrsson, H. H. Mental imagery changes multisensory

- perception. *Curr. Biol.* **23**, 1367–1372 (2013).
5. Berger, C. C. & Ehrsson, H. H. The Content of Imagined Sounds Changes Visual Motion Perception in the Cross-Bounce Illusion. *Sci. Rep.* **7**, 40123 (2017).

(2a) Second, imagery can provide a useful paradigm to investigate mental representation. However, because the nature of imagery which is covert and without any external cues, it also introduces a lot of problems that need to control. For example, how can the authors be sure that participants performed the motor imagery as instructed?

We understand the concern of the reviewer, and in fact, this concern applies to all motor imagery studies. Motor imagery is a well-established method in experimental psychology, but because of its covert nature, it is difficult to control. We believe, however, that if participants did not perform the imagery task properly, it would go against our results: we would expect no attenuation in Experiment 1 and performances similar to the baseline. Given that we observed sensory attenuation when participants imagined pressing and that this attenuation had comparable magnitude to the attenuation when participants were actually pressing, it is very reasonable to assume that the participants performed the imagery task as required.

(2b) More importantly, this study implemented a concurrent imagery-stimulation paradigm. How can the timing and dynamics of motor imagery consistent with the occurrence of external stimulation? This is important because it has been demonstrated that the prediction generated from internal forward model is very precise. In order to observe the prediction-induced suppression effect, the prediction should be well aligned with external stimulation both in terms of timing and dynamics, as well as frequency and magnitude [e.g. Houde, Nagarajan, Kekihara & Merzenich, 2002; Tian & Poeppel, 2015]. If there is any deviance between prediction and external stimulation, enhancement effects could be observed instead [e.g. Behroozman, Karvelis, Liu & Larson, 2009; Tian & Poeppel, 2015]. In the current study, an external constant force was applied to participants' left index finger, but the self-press or imagined press arguably cannot remain on a consistent level or start/end at the same time as the external force (the author can check their pressing data which is not provided). Therefore, there were very likely deviance between the prediction and external stimulation, and hence possibly created complicated suppression as well as enhancement. The authors may need to perform some follow-up studies that either have better control of performance timing and dynamics, or have a better procedure that bypass these difficulties. One suggestion is to use repetition/adaptation paradigm in combination with mental imagery (e.g. Tian & Poeppel, 2013).

We thank the reviewer for his/her comment and especially for affording us the opportunity to explain why we chose the specific design and why we believe that this design is optimal for the current experimental purposes.

Timing: When we designed this experiment, we seriously considered the importance of timing between the external tactile stimuli (left index finger) and the participants' imagined/real movement (right index finger). Previous studies of somatosensory attenuation have shown that attenuation is reduced or even eliminated when the tactile consequences of a brief movement are delayed^{6,7}. No enhancement has been previously observed in the somatosensory domain.

According to our design, in every trial, there was an auditory cue (beep), and immediately afterward, external touch was applied. The participants were instructed to imagine pressing immediately after the auditory cue and as soon as they perceived the intensity of the reference force. Recall that the participants were asked to imagine applying a force sufficiently strong to generate the reference force – thus, the participants first had to perceive the intensity of the reference force. Therefore, the touch on the left index finger was always applied before the participants started to press and, presumably, before the participants started to imagine. That is, it is safe to assume that the onsets of external touch and motor imagery did not coincide.

This asynchrony, however, is unavoidable under experimental settings with imagery since it is impossible to control *when* participants start to imagine. To specifically address this asynchrony issue when designing the experiment, we intentionally set the width of the temporal window for the reference force to 3 seconds. By doing so, we can confidently assume that within the 1st second of the reference force, the participants will have started to imagine; therefore, the duration during which the two events (imagery and external touch) overlap will be at least 2 seconds. We consider this methodological approach to be reliable to ensure temporal alignment since longer intervals increase the probability of this co-occurrence.

Dynamics: We encouraged participants to tune the force that they imagined at the very beginning of the trial so that it matched the intensity required to generate the external touch and then to remain stable for the rest of the reference force. In contrast, the reference force that participants received on their left index finger was constant from the beginning of the trial since it served as the point of reference for the participants' imaginary forces. Besides the very beginning of the trials, when participants need to perceive the reference force to decide what to imagine, we have no reason to believe that the dynamics differed for the majority of the reference force duration.

Magnitude: In each trial, it was the motor that set the magnitude of the reference force. A previous study by Bayes and Wolpert⁸ showed that – at least in the somatosensory domain – a precise match between the magnitude of the force with which the participant pressed with his/her right hand and the magnitude of the force that the participants simultaneously received on the left hand is not strictly required. However, we aimed to establish a match in terms of magnitude, which is why we asked participants to imagine pressing as much as they thought they should to match the magnitude on the left hand.

Repetition/adaptation paradigm: We thank the reviewer for his/her suggestion. Although the repetition/adaptation design described in Tian and Poeppel⁹ might be appropriate for studying the effect of imagery on subsequent auditory perception, we

did not understand how it would be adequate for current experimental purposes.

Particularly, in the paper by Tian and Poeppel⁹, the authors were interested in determining whether ‘*an internally generated representation (elicited in a mental imagery task) can act as an adaptor for a subsequent overt probe stimulus –or more colloquially, whether thought will prime perception*’. Participants were asked to (a) articulate a syllable, (b) imagine articulating a syllable, (c) hear a syllable or (d) imagine hearing a syllable (adaptor), and they were tested on their perceptions of a probe syllable heard afterward. The adaptor and the probe syllable could be the same or not. The authors measured neural responses (MEG), and they assessed whether the neural responses during the probe stimulus differed depending on the task that the participants performed before (adaptor).

We believe that the two research questions are fundamentally different (sensory attenuation versus suppression/enhancement of neural responses due to repetition of stimuli) because their underlying mechanisms seem to be different, which is why distinct methodologies are required. The sensory attenuation phenomenon is believed to rely on the predictions of the forward models that are driven by the efference copy. That is, to study the presence of sensory attenuation, *we require the presence of the efference copy at the same time that the stimulus is presented*, which means that the repetition/adaptation design is not applicable for studying sensory attenuation. Even very small delays between the presence of the efference copy and that of the tactile stimulus can eliminate sensory attenuation⁷, like those employed in the paper by Tian and Poeppel. Using the repetition/adaptation paradigm to study attenuation during motor imagery would mean that we should ask participants to imagine a press and then assess how they perceive a touch that comes later. Such a design would have detrimental effects for causality because the touch cannot be perceived as a consequence of the pressing since (a) the occurrences of the two events are different in time and there is no overlap at any time and (b) the magnitude of the force that the participants imagine is completely unrelated to the force that they will receive afterwards – participants cannot imagine what they *will* perceive (if they have not perceived it before).

However, crucially, the results of using such a design would be uninformative to the research question at hand: if sensory attenuation is not observed (as expected), we could not argue that motor imagery does not involve predictions of the forward models since there is no efference copy at that time to drive the forward models. Thus, we believe that this alternative design is not adequate for the purpose of the present study, which is to study whether forward models predict the consequences of the imagined movement.

References

6. Blakemore, S. J., Frith, C. D. & Wolpert, D. M. Spatio-temporal prediction modulates the perception of self-produced stimuli. *J. Cogn. Neurosci.* **11**, 551–559 (1999).
7. Bays, P. M., Wolpert, D. M. & Flanagan, J. R. Perception of the consequences of self-action is temporally tuned and event driven. *Curr. Biol.* **15**, 1125–1128 (2005).
8. Bays, P. M. & Wolpert, D. M. in *Sensorimotor Foundations of Higher Cognition* (eds. Haggard, E. P., Rosetti, Y. & Kawato, M.) **22**, 339–358

- (Oxford University Press, 2008).
9. Tian, X. & Poeppel, D. The Effect of Imagination on Stimulation: The Functional Specificity of Efference Copies in Speech Processing. *J. Cogn. Neurosci.* **25**, 1020–1036 (2013).

(3) There is one particular setting of experiments that may seriously undermined the validity of this study. In this current study, the probe to the left index finger is independent from the force exerted from the right hand, and participants were aware that the two forces were unrelated (Pg. 11). This setting violates the critical assumption about internal forward model of sensory prediction, which is that the action is the cause of the sensory consequence. This unrelated setting between the button-press and external force makes the paradigm and results about internal forward model of sensory prediction in serious doubts.

We understand the concerns of the reviewer. However, in our opinion, there is no doubt that our setup maintains the perceived causality between the pressing and the felt touch while at the same time making it technically feasible to psychophysically test sensory attenuation during motor imagery. Below, we discuss this issue in greater detail.

Do we break the causality? Participants were asked to press/imagine pressing their right index finger as much as they thought that was necessary to generate the touch applied on their left index finger. In other words, the task effectively simulates direct self-touch. Although the participants were verbally informed before the experiment that the pressure and the touch were unrelated (so that they would not think that there was something wrong with our equipment if they accidentally changed their force slightly with their pressing right index finger and did not sense the corresponding perceived force with the receiving left index finger), we asked them to generate the force that they believed would fit the touch applied by the motor as if this touch were the direct sensory consequence of the imagined or executed movement. Therefore, although there was no true causality in a physical sense, we are confident that our participants perceived a causal relationship between their (executed or imagined) right index finger action and the touch felt on the left index finger. Importantly, if they did not perceive this causal self-touch, it would undermine our results because the perceived forces would then not be attenuated. Thus, a putatively reduced sense of causality introduced by our setup, compared to direct self-touch between fingers, cannot explain our results. In fact we observed robust sensory attenuation in the press and imagine conditions, so we believe the present setup simulates direct physical self-touch well. We have now modified the text on Lines 432-439 to explain this point better.

Why did we choose this design? To psychophysically study sensory attenuation, we used the force-matching task, i.e., a well-established paradigm in experimental psychology to study sensory attenuation^{8,10-14}. For the task, we had to provide several levels of our stimulus (reference force) and record the slider responses of the participants. To provide a (different) level of the reference force (i.e., 1 N, 1.5 N, 2 N,

2.5 N etc.) in each trial, we had to provide the forces externally (with the motor), which is the only way to control for our independent variable and study sensory attenuation with both real and imagined pressing.

Alternatively, we could have linked the touch that participants received on their left index finger with the pressure that they generated with their right index finger. However, the imagery condition would have been *technically impossible* to perform in this scenario (how would the imagined pressure control the touch applied on the left index finger?). However, even if we ignored the imagery condition and focused on the press condition, the participants would be required to learn before the experiment to press a fixed amount of force that differed in each trial. Based on our experience, such an outcome is technically very difficult to achieve. We previously conducted a pilot experiment in which the participants were asked to learn to press 1 N or 2 N or 3 N under three different experimental conditions (a much easier task than learning different forces for each trial). We observed that it was extremely difficult for people to remember to apply the same force (without any feedback) for 35 trials and not to become distracted by the simultaneous touches on their left index finger. Therefore, we believe that it would be nearly impossible to train people to press different levels of forces (1 N, 1.5 N, 2 N, 2.5 N, 3 N) under the same condition, and this design would yield very unreliable results.

References

8. Bays, P. M. & Wolpert, D. M. in *Sensorimotor Foundations of Higher Cognition* (eds. Haggard, E. P., Rosetti, Y. & Kawato, M.) **22**, 339–358 (Oxford University Press, 2008).
10. Shergill, S. S., Bays, P. M., Frith, C. D. & Wolpert, D. M. Two eyes for an eye: the neuroscience of force escalation. *Science* **301**, 187 (2003).
11. Shergill, S. S., Samson, G., Bays, P. M., Frith, C. D. & Wolpert, D. M. Evidence for sensory prediction deficits in schizophrenia. *Am. J. Psychiatry* **162**, 2384–2386 (2005).
12. Walsh, L. D., Taylor, J. L. & Gandevia, S. C. Overestimation of force during matching of externally generated forces. *J. Physiol.* **589**, 547–557 (2011).
13. Kilteni, K. & Ehrsson, H. H. Sensorimotor predictions and tool use: Hand-held tools attenuate self-touch. *Cognition* **165**, 1–9 (2017).
14. Kilteni, K. & Ehrsson, H. H. Body ownership determines the attenuation of self-generated tactile sensations. *Proc. Natl. Acad. Sci.* 201703347 (2017). doi:10.1073/PNAS.1703347114

Minor points:

(4) I would not call the aim of this study ‘computational equivalence’ as compared to ‘functional equivalence’, because the observed imagery-induced perceptual suppression is the function of internal forward model – predicting the perceptual consequence of actions.

We understand the possible confusion that these terms might bring. By ‘functional equivalence’, we do not strictly refer to the ‘function’ of the internal forward model. Instead, we refer to the term widely used in the motor imagery literature^{15–19} to describe the concept that imagined and overt movements involve similar

representations, neural structures and processes (e.g., motor intentions, motor planning and preparation) but differ in the stage of motor execution. We believe that using this well-known term would facilitate the reading of our manuscript. Therefore, if possible, we would prefer not to change our term ‘computational equivalence’ because we believe that it describes well our hypothesis and findings, and it emphasizes that we sought very specific patterns of behavioral results that precisely followed the predictive computations of the internal models, as opposed to mere general behavioral similarities as captured by the broader term ‘functional equivalence’.

References

15. Di Rienzo, F. *et al.* Online and Offline Performance Gains Following Motor Imagery Practice: A Comprehensive Review of Behavioral and Neuroimaging Studies. *Front. Hum. Neurosci.* **10**, 315 (2016).
16. Grezes, J. & Decety, J. Functional Anatomy of Execution, Mental Simulation, Observation, and Verb Generation of Actions: A Meta-Analysis. *Hum. Brain Mapp.* **12**, 1–19 (2001).
17. Moran, A., Guillot, A., MacIntyre, T. & Collet, C. Re-imagining motor imagery: Building bridges between cognitive neuroscience and sport psychology. *British Journal of Psychology* **103**, 224–247 (2012).
18. Decety, J. & Jeannerod, M. Mentally simulated movements in virtual reality: does Fitts’s law hold in motor imagery? *Behav. Brain Res.* **72**, 127–134 (1995).
19. Jeannerod, M. The representing brain: Neural correlates of motor intention and imagery. *Behav. Brain Sci.* **17**, 187 (1994).

(5) Why did the base condition show negative attenuation in experiment 1 but no attenuation in experiment 2? Should they be the same since they are rest conditions only differ in the position of rest?

The *base* condition is a baseline condition that assesses participants’ force perception. Therefore, variability between groups of participants is expected. We performed a two-sample t-test to determine whether there was any difference between the baselines of the two groups (the distributions were normal according to the Shapiro-Wilk test). There was no difference between the baselines: $t(17.56) = 0.75$, $p = 0.460$, $CI = [-0.201, 0.424]$.

(6) Why did the authors choose a between subject design? Though the authors used base conditions to normalize the attenuation effect in imagery condition, it can still introduce some confounds. For example, it may be that base and imagery have complicated interaction caused by the distance of button placed. Therefore, a within subject design would be better.

We opted for a between-groups design only for practical reasons. Each experiment required approximately one hour. In addition, some trials had to be rejected and performed again because of visible muscular activity when the participants should have been relaxed. That is, performing a within-groups experiment (with six

conditions) would have required two hours at least, which we deemed too long for keeping the participants focused on the task, based on our previous experiences in the lab.

In our revised manuscript, we report the effect sizes for all statistical tests. We would like to emphasize that our main finding of attenuation during imagery comes from a within-subjects comparison in Experiment 1 (*imagine* vs *base*: $p = 0.018$, Hedges' $g_{av} = 0.551$). This difference was absent in Experiment 2 (*imagine_{far}* vs *base_{far}*: $p = 0.345$, Hedges' $g_{av} = 0.175$). Then, we performed a between-groups comparison to directly compare the effects of distance, and we found a significant difference with a large effect size: $p = 0.020$, Cohen's $d_s = 1.024$. In other words, although a between-subjects design reduces the statistical power compared to a within-groups design, our effect is large.

Finally, we do not have any particular reason to consider that a complicated interaction between *base* and *imagine* driven by distance is likely. The effect of distance on sensory attenuation has been previously shown by our group and others using within-subjects designs^{8,13,14}.

References

8. Bays, P. M. & Wolpert, D. M. in *Sensorimotor Foundations of Higher Cognition* (eds. Haggard, E. P., Rosetti, Y. & Kawato, M.) **22**, 339–358 (Oxford University Press, 2008).
13. Kilteni, K. & Ehrsson, H. H. Sensorimotor predictions and tool use: Hand-held tools attenuate self-touch. *Cognition* **165**, 1–9 (2017).
14. Kilteni, K. & Ehrsson, H. H. Body ownership determines the attenuation of self-generated tactile sensations. *Proc. Natl. Acad. Sci.* 201703347 (2017). doi:10.1073/PNAS.1703347114

Reviewer #2 (Remarks to the Author):

(1) One important remark concerns the fact that the authors' main conclusions are based on the absence of a statistical difference between two conditions, namely the 'press' and 'imagine' conditions. I think this poses problems for two reasons: For one, the authors bump into an interpretation problem that is reminiscent of the 'reverse inference' problem in the fMRI literature, which is in my opinion problematic. Based on the presence of similar behavioral/neural effects in two conditions, one cannot argue that identical cognitive mechanisms are necessarily underlying the effects. It might indeed be the case that a similar cognitive mechanism is underlying the attenuation of touch within both conditions, but this is not necessarily the case, logically speaking. It might also be the case that very distinct cognitive mechanisms produce a similar behavioral effect. How would the authors respond to this? Second, the authors do not provide measures that directly test and quantify the absence of a statistical difference (e.g., Bayesian tests or equivalence test). While the sample size of the experiments is relatively small (12

individuals in each experiment), subtle differences between the two conditions (that might have been present when a larger sample would have been used) might not have been detected in the current experiments.

We thank the reviewer for his/her positive comments.

First, we would like to emphasize that our conclusions were based on *significant differences* between conditions and not on the absence of a statistical difference. On Lines 165-167, after reporting the main effect of conditions in ANOVA, we report a significant difference between *imagine* and *base* conditions ($p = 0.018$), which shows that the reference force on the left index finger was attenuated when participants imagined a movement that had tactile consequences on the left index finger, which is the critical result that supports the main conclusion of the paper and, as can be seen, is based on a significant difference between conditions. The size of this effect with only 12 participants is medium: Hedges' $g_{av} = 0.551$. Based on this statistically significant result, we conclude that the sensory consequences of the imagined movement are predicted because the touch is attenuated.

In addition to this outcome, we found that the difference between the imagery and baseline conditions is absent in the control experiment (Experiment 2, Lines 244-247, $p = 0.345$), in agreement with our hypothesis. The effect of the difference (if any) was small (Hedges' $g_{av} = 0.175$). Based on the absence of any significance, we argue that there is probably no attenuation in this control condition, further strengthening our main conclusion. However, because we wanted to avoid any 'reverse inference', we directly compared the imagery-related effects between groups (Lines 269-274) and did observe significantly stronger attenuation in the critical imagery condition in Experiment 1, compared to the imagery control condition in Experiment 2 ($p = 0.020$; with a large effect size Cohen's $d_s = 1.024$). This significant difference again supports the main conclusion of the paper. In summary and to be absolutely clear, our key conclusion is supported by two statistically significant differences between conditions.

Nevertheless, the reviewer is correct in that we use the expression 'the same magnitude' twice in the text (in the Abstract and on Page 8), based on the absence of the statistical difference between *press* and *imagine* in Experiment 1 ($p = 0.908$). In contrast, in other parts of the text, we used the word 'similar', which is more appropriate given that we are referring to a non-statistically significant difference. Accordingly, we have now edited the text, replacing the word 'same' with 'similar' or 'almost identical' to avoid any overstatements.

- Line 24: is of similar magnitude
- Line 291: of comparable magnitude

It is interesting to note that the size of any effect between *press* and *imagine* is very small: Hedges' $g_{av} = 0.019$. This finding indicates that such a very small effect would require a very large sample to be detected. Moreover, we followed the suggestion of the reviewer, and we also calculated the Bayes factor for this comparison (R package *BayesFactor* 0.9.12-2): $BF_{10} = 0.29 \pm 0.02\%$. This finding suggests that the observed

data are 3.46 times more likely to have occurred under the null hypothesis (*press* and *imagine* are the same) than under the alternative hypothesis. Furthermore, we should also point out that our main conclusion does not depend on whether or not the difference between the *press* and *imagine* conditions is significant or not. We know from many previous studies that mental imagery is sometimes less vivid than real perception and can produce weaker behavioral effects (e.g., ⁴). However, even if we had found significantly weaker attenuation in the imagery condition (which we in fact did not), our main conclusion described above about imagery engaging the forward models would still be correct. Thus, let us again underscore that we are not basing our main conclusion on the non-significant difference between the *press* and *imagine* conditions.

Finally, we consider that our deduction that ‘since both tasks (*press* and *imagine*) produce attenuation, then a common mechanism underlies both tasks’ is very reasonable. From a theoretical perspective, we can see no alternative hypothesis for how attenuation would be produced in the imagery conditions without forward models and efference copies. Moreover, many experimental studies of motor imagery have shown that imagined movements share several similarities with overt movements in different aspects, such as duration, trade-off between duration and task difficulty, physiological activation and neural networks (see ^{16,20} for reviews). Finding precise and systematic behavioral similarities between imagery and execution of motor tasks is strongly suggestive of a shared underlying mechanism in terms of forward models. In addition, a common underlying mechanism would be computationally less expensive than different mechanisms for covert and overt movements. However, we agree with the reviewer that we cannot exclude the possibility of different mechanisms governing overt and imaginary movements. In our manuscript we have attempted to avoid any overstatements by simply stating that we propose that the most likely explanation is that the mechanism responsible for sensory attenuation in real and imagined movements is the same (Lines 309-310).

References

4. Berger, C. C. & Ehrsson, H. H. Mental imagery changes multisensory perception. *Curr. Biol.* **23**, 1367–1372 (2013).
16. Grezes, J. & Decety, J. Functional Anatomy of Execution, Mental Simulation, Observation, and Verb Generation of Actions: A Meta-Analysis. *Hum. Brain Mapp.* **12**, 1–19 (2001).
20. Héту, S. *et al.* The neural network of motor imagery: An ALE meta-analysis. *Neurosci. Biobehav. Rev.* **37**, 930–949 (2013).

(2) Another major remark is related to the problem of specificity of the presumed efference copy. If the sensory attenuation that was observed in the imagine conditions is truly consequent to efference copies, another imagined movement that has different tactile consequences should not produce touch attenuation. I wonder, in other words, whether the attenuation is due to the fact that participants imagine the specific movement they are asked to imagine (i.e. pressing the right index finger against the sensor), or to the fact that they imagine touch per se (which could have been produced by any movement). While the authors would like to argue that the

attenuation in the imagine condition is consequent to efference copies, this is an important question to ask. To solve this issue, I am afraid an additional control experiment will be needed. I suggest to manipulate the congruency of the imagined action (e.g. index finger or middle finger movement) and the felt touch (i.e., felt touch on index finger or middle finger; for a similar discussion on movement specificity in observed finger movements and felt touch, see Deschrijver, Wiersema & Brass, 2015; 2016). Only in the case where the sensory attenuation would be observed in congruent trials but not in incongruent trials, the authors would be able to effectively argue that tactile consequences of the specific imagined movement may underlie the attenuation effects.

We thank the reviewer for this comment. We believe that there has been a misunderstanding, and we apologize for not presenting sufficiently clearly our control experiment (Experiment 2) or its purpose in the previous version of the manuscript.

First, we are in strong agreement with the reviewer, which is exactly why we conducted our control experiment: to show that an imagined action of the right index finger that is incongruent with touch on the left index finger does not produce attenuation. We hypothesized exactly what the reviewer suggests: “if the sensory attenuation that was observed in the imagine condition is truly consequent to efference copies, another imagined movement that has different tactile consequences should not produce touch attenuation”. For this reason, we designed a control experiment consisting of exactly the same task and instructions as in Experiment 1 but differs in the sense that the touch on the left index finger is not predicted by the imagined action of the right index finger. By placing the hands far apart (25 cm), the forward model does not predict a consequence of the right hand’s movement to the left hand; thus, the touch on the left index finger is not attenuated, which is exactly what our control experiment shows: no significant differences between the *imagine* and *baseline* conditions. This control condition, in which a distance between the hands is introduced, has been previously shown to eliminate sensory attenuation,^{8,13,14} and for the present purposes, it is an ideal control condition since it matches all of the characteristics of the experimental condition except the distance and attenuation. Thus, by comparing the key experimental condition to this control condition, we match, and therefore eliminate, any effects related to performing motor imagery in general because we are comparing two very similar motor imagery tasks (same finger, same pressing action), any effects related to somatosensory imagery (same imagined finger movement and imagined force at the fingertip), and any effects related to attention or cognitive efforts (same demands in both conditions).

Second, we did not understand why using a different finger would make the trials incongruent, as the reviewer suggested. Even if participants used their middle finger for the imaginary pressing, we would still expect the same result because the movement of the middle finger would predict the touch on the index finger placed below it; i.e., the pressing and the receiving fingers do not need to be the same fingers.

Thus, we believe that our experimental design achieves exactly what the reviewer is

asking for by demonstrating significantly stronger attenuation in the ‘congruent trials’ (Experiment 1 – imagined movement of the right index finger predicts touch on the left index finger below it), compared to the ‘incongruent trials’ (Experiment 2 – imagined movement of the right index finger does not predict touch on the left index finger 25 cm away). Therefore, the efference copy of the imagined movement matters. We have added new text to better present the rationale for our control experiment (Lines 192-212 and 255-260), which was poorly explained in the original version. We hope that we have now convinced the reviewer that our control experiment controls for the potentially confounding factors that he/she mentioned.

References

8. Bays, P. M. & Wolpert, D. M. in *Sensorimotor Foundations of Higher Cognition* (eds. Haggard, E. P., Rosetti, Y. & Kawato, M.) **22**, 339–358 (Oxford University Press, 2008).
13. Kilteni, K. & Ehrsson, H. H. Sensorimotor predictions and tool use: Hand-held tools attenuate self-touch. *Cognition* **165**, 1–9 (2017).
14. Kilteni, K. & Ehrsson, H. H. Body ownership determines the attenuation of self-generated tactile sensations. *Proc. Natl. Acad. Sci.* 201703347 (2017). doi:10.1073/PNAS.1703347114

(3) On a similar note, one could wonder to which extent participants imagined the touch itself that followed the imagined movement, next to the movement alone. While the authors did not provide a post-Likert measure for the vividness of touch imagery (e.g., “I could ‘feel’ how the imagined movement pressed my finger.” Or “I found that when I imagined moving my finger, the touch seemed clear and vivid, almost as if the touch were real”), the likelihood of this potential alternative explanation to interpret the data is difficult to assess. However, if true, one would not need to infer an explanation in terms of efference copies: the extent to which the participant imagined the touch consequent to the action would explain the observed effects. For this reason, I would suggest to include a post measure of imagined vividness of touch imagery in the control experiment: if this measure is found unrelated to the attenuation effects, an interpretation in terms of efference copies may be more likely (as an efference copy should theoretically be present anyway following the imagined movement).

We thank the reviewer for this very interesting comment on tactile imagery. Because we did not explain this topic properly in the previous version of the manuscript, the important point that tactile and proprioceptive imagery were perfectly controlled for by the imagery control condition introduced in Experiment 2 was probably missed. In our study, we instructed the participants to imagine pressing their right index finger against the sensor and more specifically, to imagine the feeling of force and muscle contraction in the right index finger associated with this action. This movement (if executed) includes tactile feedback from the sensor of the right index finger, so the participants most likely imagined the touch on their right index finger as well. Importantly, however, this aspect of the motor imagery was exactly the same as in the imagery control condition in Experiment 2. In this second experiment, the imagery condition included the exact same instructions for the imaginary movement.

Therefore, both imagery conditions should include the same tactile imagery as part of the imagined pressing action. As described in the manuscript, the only difference between the imagery conditions in Experiments 1 and 2 was the distance between the hands, with the large distance in Experiment 2 eliminating sensory attenuation (as is well known from previous studies^{8,13,14}). That participants attenuated the touch on the left index finger only in Experiment 1 and not in Experiment 2 excludes the possibility that touch imagery can explain our results. Having said that, we can also add that, from a theoretical perspective, we see no reason why somatosensory imagery should produce sensory attenuation. We thank the reviewer for bringing this matter to our attention, and in the new version of the manuscript, we have now clarified that Experiment 2 controlled for these important potentially confounding factors, including somatosensory imagery (Line 257).

References

8. Bays, P. M. & Wolpert, D. M. in *Sensorimotor Foundations of Higher Cognition* (eds. Haggard, E. P., Rosetti, Y. & Kawato, M.) **22**, 339–358 (Oxford University Press, 2008).
13. Kilteni, K. & Ehrsson, H. H. Sensorimotor predictions and tool use: Hand-held tools attenuate self-touch. *Cognition* **165**, 1–9 (2017).
14. Kilteni, K. & Ehrsson, H. H. Body ownership determines the attenuation of self-generated tactile sensations. *Proc. Natl. Acad. Sci.* 201703347 (2017). doi:10.1073/PNAS.1703347114

Additionally, I have some more minor comments:

(4) What are the effect sizes of the findings? I suggest to add these to the manuscript.

We apologize for not reporting effect sizes earlier. Effect sizes were estimated with the Hedges g_{av} for paired t-tests and with Cohen's d_s for unpaired t-tests, as suggested by²¹. For the Mann-Whitney U -test and Wilcoxon's signed-rank test, effect sizes were calculated as $r = Z / \sqrt{N}$, where N is the total sample size of the given test (i.e., $N = 24$). We have now added the effect sizes for all comparisons, and we have also added text to the Methods section to describe how we computed them (Lines 533-536).

As can be seen, the effect size for the comparison between *imagine* and *base* in Experiment 1 is of medium size ($g_{av} = 0.551$). The same comparison in the control experiment has a quite small effect ($g_{av} = 0.175$), while the between-experiments comparison yields a large effect: Cohen's $d_s = 1.024$.

Reference

21. Lakens, D. Calculating and reporting effect sizes to facilitate cumulative science: A practical primer for t-tests and ANOVAs. *Front. Psychol.* **4**, (2013).

(5) It is not entirely clear to the reader what is meant exactly with terms like 'computational mechanisms' or 'predictive computational units'. I suggest to elaborate on this.

With regard to the term ‘predictive computational units’, we refer to the forward models. This term is quite commonly used in the computational neuroscience literature when scientists want to introduce and explain forward models to a broader audience, which is why we used it. Additionally, when we introduce this term in the manuscript, it should be clear to the reader that we refer to the forward models. We have now changed the text on Lines 41-44 to be more precise.

With the term “mechanisms”, most people think of “physical mechanisms”, and in neuroscience, this thought of course translates to the action potentials of neurons. However, in psychology for example, researchers often talk about “psychological mechanisms” when they want to explain causal relationships between psychological processes and behavior. In a similar vein, it is not uncommon to see researchers in computational neuroscience use the term “computational mechanisms” (e.g., Franklin and Wolpert 2011 Neuron, ‘Computational mechanisms of sensorimotor control’) when they explain how behavior could be produced by computational models. Since we have now clarified what we mean by “predictive computational units” in the paragraph above (Lines 41-44), it should be clear what we mean by “computational mechanisms” in Lines 76-77.

(6) The authors use their model to make theoretical claims about the role of efference copy for imagined movements. However, it should be noted that the findings are based on movements that involve a clear tactile consequence (i.e. a pressing movement). I think the theoretical implications should therefore be limited to efference copies of these specific types of movements, rather than to efference copies of imagined movement per se.

We thank the reviewer for this comment. We acknowledge that not all movements produce tactile feedback, but we believe that our results have general implications beyond somatosensation. First, the force-matching task and the sensory attenuation studies are among the most used model systems to investigate sensorimotor control, so we believe that we should not restrict our conclusions to somatosensation. Second, efference copies are a central concept in all theories of sensorimotor control, and as far as we are aware, the basic principles are not different for different types of movements, so the present results should generalize well beyond the self-touch paradigm in our opinion. Third, we would like to emphasize that participants imagined moving their right hand, while the perception of touch on their left hand was attenuated, indicating that the forward models first had to predict the visual and proprioceptive consequences of the right hand that were expected after the (imaginary) movement (i.e., its new position). Since the predicted position of the left hand is sufficiently close to the predicted state of the right hand, tactile consequences are predicted for the left hand. In other words, the tactile predictions arise as a consequence of efference copies and visual and proprioceptive predictions about future hand states. We therefore believe that our conclusions concern fundamental principles of sensorimotor control and efference copies rather than processes specific to our task.

(7) It is unclear to me why the authors blindfolded participants in the imagine conditions but not in the baseline and movement conditions. This leads to a difference between conditions in the sense that that behavioral measures in the imagine condition are based on unimodal information (tactile), whereas those of the movement/baseline condition are based on multimodal information (visuotactile). Though I do understand that even in these multimodal conditions, participants did not visually perceive their hands, this does account for a modality difference over conditions in the methodological design. In addition, the unimodal findings (based on the tactile sense only) in the imagine condition are thus compared to a multimodal baseline condition (which was also visuotactile). Could the authors comment on why this methodological choice was made? While it is not entirely clear to me to which extent this choice could have influenced or biased the results, I think this should be noted to the attentive reader (and be avoided in future experiments)

We blindfolded the participants in the motor imagery conditions of the two experiments because it rendered the imagery task easier by suppressing visual distractions. However, the reviewer is probably correct; it would be methodologically better to have the participants blindfolded under all conditions of both experiments. However, in previous experiments with the force-matching task, the participants were not blindfolded (e.g., ^{8,10,13,14}), which is why we used the same approach in our motor execution and baseline measurements of sensory attenuation. We do not expect that blindfolding, having the eyes open or eyes closed, or having a fixation point or not matters for the sensory attenuation phenomenon in the force-matching task. In the force-matching paradigm, vision is not informative about the task and therefore should not influence performance. However, in direct response to the reviewer's concerns, we would like to emphasize that the subjects were blindfolded under *both* imagery conditions in Experiments 1 and 2, so any putative effect related to blindfolding is perfectly matched when comparing these conditions. Thus, the effects of blindfolding cannot explain the significant difference in sensory attenuation that we observed between these two imagery conditions.

References

8. Bays, P. M. & Wolpert, D. M. in *Sensorimotor Foundations of Higher Cognition* (eds. Haggard, E. P., Rosetti, Y. & Kawato, M.) **22**, 339–358 (Oxford University Press, 2008).
10. Shergill, S. S., Bays, P. M., Frith, C. D. & Wolpert, D. M. Two eyes for an eye: the neuroscience of force escalation. *Science* **301**, 187 (2003).
13. Kilteni, K. & Ehrsson, H. H. Sensorimotor predictions and tool use: Hand-held tools attenuate self-touch. *Cognition* **165**, 1–9 (2017).
14. Kilteni, K. & Ehrsson, H. H. Body ownership determines the attenuation of self-generated tactile sensations. *Proc. Natl. Acad. Sci.* 201703347 (2017). doi:10.1073/PNAS.1703347114

Reviewer #3 (Remarks to the Author):

(1) This sentence in the intro is a bit contradictory with respect to MI, as it leaves the reader unclear as to whether MI activity is expected or not during imagery. “Neuroimaging studies further showed that motor imagery activates a set of non-primary motor areas, parietal areas, and cerebellar regions that partially overlaps with the brain network that is activated during motor execution^{9,25} (for reviews, see^{6,7}) and that motor imagery of different effectors activates the corresponding sections of the somatotopically organized motor cortex⁵.”

We thank the reviewer for his/her positive comments.

We agree with the reviewer, and we accordingly have changed the phrase so that it does not sound contradictory (Lines 34-38).

(2) Line 179: “reference forces”

We thank the reviewer for observing this typo. We have corrected it (Line 180).

(3) Line 195: “in exactly the same conditions”

We thank the reviewer for observing the error. We have corrected it (Line 202).

(4) Figure 3C: can’t see the median...

We have increased the thickness of the median line in Figure 3C. To be consistent, we have also done the same for the boxplots in Figure 3B.

All references

1. Tian, X. & Poeppel, D. Mental imagery of speech and movement implicates the dynamics of internal forward models. *Front. Psychol.* (2010). doi:10.3389/fpsyg.2010.00166
2. Tian, X., Zarate, J. M. & Poeppel, D. Mental imagery of speech implicates two mechanisms of perceptual reactivation. *Cortex* **77**, 1–12 (2016).
3. Scott, M. Corollary Discharge Provides the Sensory Content of Inner Speech. *Psychol. Sci.* **24**, 1824–1830 (2013).
4. Berger, C. C. & Ehrsson, H. H. Mental imagery changes multisensory perception. *Curr. Biol.* **23**, 1367–1372 (2013).
5. Berger, C. C. & Ehrsson, H. H. The Content of Imagined Sounds Changes Visual Motion Perception in the Cross-Bounce Illusion. *Sci. Rep.* **7**, 40123 (2017).
6. Blakemore, S. J., Frith, C. D. & Wolpert, D. M. Spatio-temporal prediction modulates the perception of self-produced stimuli. *J. Cogn. Neurosci.* **11**, 551–559 (1999).

7. Bays, P. M., Wolpert, D. M. & Flanagan, J. R. Perception of the consequences of self-action is temporally tuned and event driven. *Curr. Biol.* **15**, 1125–1128 (2005).
8. Bays, P. M. & Wolpert, D. M. in *Sensorimotor Foundations of Higher Cognition* (eds. Haggard, E. P., Rosetti, Y. & Kawato, M.) **22**, 339–358 (Oxford University Press, 2008).
9. Tian, X. & Poeppel, D. The Effect of Imagination on Stimulation: The Functional Specificity of Efference Copies in Speech Processing. *J. Cogn. Neurosci.* **25**, 1020–1036 (2013).
10. Shergill, S. S., Bays, P. M., Frith, C. D. & Wolpert, D. M. Two eyes for an eye: the neuroscience of force escalation. *Science* **301**, 187 (2003).
11. Shergill, S. S., Samson, G., Bays, P. M., Frith, C. D. & Wolpert, D. M. Evidence for sensory prediction deficits in schizophrenia. *Am. J. Psychiatry* **162**, 2384–2386 (2005).
12. Walsh, L. D., Taylor, J. L. & Gandevia, S. C. Overestimation of force during matching of externally generated forces. *J. Physiol.* **589**, 547–557 (2011).
13. Kilteni, K. & Ehrsson, H. H. Sensorimotor predictions and tool use: Hand-held tools attenuate self-touch. *Cognition* **165**, 1–9 (2017).
14. Kilteni, K. & Ehrsson, H. H. Body ownership determines the attenuation of self-generated tactile sensations. *Proc. Natl. Acad. Sci.* 201703347 (2017). doi:10.1073/PNAS.1703347114
15. Di Rienzo, F. *et al.* Online and Offline Performance Gains Following Motor Imagery Practice: A Comprehensive Review of Behavioral and Neuroimaging Studies. *Front. Hum. Neurosci.* **10**, 315 (2016).
16. Grezes, J. & Decety, J. Functional Anatomy of Execution, Mental Simulation, Observation, and Verb Generation of Actions: A Meta-Analysis. *Hum. Brain Mapp.* **12**, 1–19 (2001).
17. Moran, A., Guillot, A., MacIntyre, T. & Collet, C. Re-imagining motor imagery: Building bridges between cognitive neuroscience and sport psychology. *British Journal of Psychology* **103**, 224–247 (2012).
18. Decety, J. & Jeannerod, M. Mentally simulated movements in virtual reality: does Fitts's law hold in motor imagery? *Behav. Brain Res.* **72**, 127–134 (1995).
19. Jeannerod, M. The representing brain: Neural correlates of motor intention and imagery. *Behav. Brain Sci.* **17**, 187 (1994).
20. Héту, S. *et al.* The neural network of motor imagery: An ALE meta-analysis. *Neurosci. Biobehav. Rev.* **37**, 930–949 (2013).
21. Lakens, D. Calculating and reporting effect sizes to facilitate cumulative science: A practical primer for t-tests and ANOVAs. *Front. Psychol.* **4**, (2013).

Reviewers' comments:

Reviewer #1 (Remarks to the Author):

I appreciate the authors' efforts in revision, which I believe make this manuscript stronger. I also sympathize the authors' position because of the difficulties of using mental imagery paradigms. However, there are still some questions that need to be addressed. I believe that investigating these questions can further significantly strengthen this study both theoretically and empirically.

My major concern is still the timing, causal relation and specific mechanism of efference copy in motor imagery. Supposed as the authors assumed that the verbal instruction and their specific designs could maintain the causality of self-action, it is subject to additional confounds such as divided attention, because presumably participants were doing two tasks at once – imagining pressing to a desired force level with one hand while evaluating the tactile consequence in another hand. Numerous studies has shown that dividing attention can cause decreases in perceiving stimulus intensity or even completely remove the sensation. Hence, the concurrent mental imagery and perceptual evaluation could also cause the observed attenuation of tactile sensation.

Moreover, in the authors' responses, they raised a rather interesting mechanistic hypothesis about the necessity of co-occurrence of efference copy and external stimulation. They assume that the external stimulus must arrive during the time of efference copy in order to have the attenuation effects. It implicitly assumes that the function of efference copy is online inhibition. Alternatively, reasoning from the quasi-perceptual experience of mental imagery, there should be a similar perceptual representation established during imagery. As long as the representation is in a similar format as perception, mental imagery should be able to create memory trace that influence on perception even after it ends. In fact, some mental imagery studies have found adaptation effects in both visual (e.g. Ganis & Schendan, 2008 Neuroimage Visual mental imagery and perception produce opposite adaptation effects on early brain potentials; Wu et al., 2012 Frontiers The effects of visual imagery on face identification: an ERP study) and auditory domains (Scott, 2013 Psych Sci). If the authors did not observe the adaptation effect in sensorimotor domain, it could also be interesting as it suggesting the specific time constant of efference copy for somatosensory and motor imagery. So again, for controlling imagery timing, attention and other confounds, as well as providing evidence for further functional aspects of efference copy, I strongly recommend the authors run an adaptation version of their experiment by simply modifying the timing of their procedure.

Reviewer #2 (Remarks to the Author):

The authors did an excellent job responding to my review. With patience and clarity, they addressed everything I brought to the table. With the exception of one small point below, I have no further remarks.

Minor comments:

1. I would suggest the authors to additionally include at least the following lines to the manuscript, which they they provided in response to one of my comments.

"We consider that our deduction that 'since both tasks (press and imagine) produce attenuation, then a common mechanism underlies both tasks' is very reasonable. From a theoretical perspective, we can see no alternative hypothesis for how attenuation would be produced in the imagery conditions without forward models and efference copies. In addition, a common underlying mechanism would be computationally less expensive than different mechanisms for covert and overt movements."

Reviewers' comments:

Reviewer #1 (Remarks to the Author):

My major concern is still the timing, causal relation and specific mechanism of efference copy in motor imagery. Supposed as the authors assumed that the verbal instruction and their specific designs could maintain the causality of self-action, it is subject to additional confounds such as divided attention, because presumably participants were doing two tasks at once – imagining pressing to a desired force level with one hand while evaluating the tactile consequence in another hand. Numerous studies has shown that dividing attention can cause decreases in perceiving stimulus intensity or even completely remove the sensation. Hence, the concurrent mental imagery and perceptual evaluation could also cause the observed attenuation of tactile sensation.

We thank the reviewer for his/her positive comments.

With respect to the reviewer's concern about the divided attention, we agree that this could be a confounding factor that produces the observed attenuation during motor imagery in Experiment 1. As stated in our manuscript (Pages 5-6, Lines 207-225), this was exactly the reason why we conducted Experiment 2: to rule out that just performing two tasks simultaneously produces the previously observed attenuation. In Experiment 2, the conditions and instructions were identical to those in Experiment 1, with the only difference being the distance between the two hands. In other words, the participants in Experiment 2 had to perform a dual task (imagining moving the right index finger and feeling the touch on the left index finger) as did the participants in Experiment 1. There are two hypotheses: If divided attention is the cause of the somatosensory attenuation observed in Experiment 1, we should observe the same pattern of responses in Experiment 2. Alternatively, if attenuation is driven by the forward models predicting the touch only when the two index fingers are likely to be in contact (Experiment 1), then we should not observe any attenuation in Experiment 2. The data from Experiment 2 clearly support the forward models hypothesis, since we did not observe any attenuation despite the fact that the participants had to perform the same dual task with the same attentional requirements.

We have now added text to our manuscript to explicitly refer to the divided attention hypothesis and to explicitly state that we controlled for this variable by performing Experiment 2 (Pages 5-6, Lines 207-225).

Moreover, in the authors' responses, they raised a rather interesting mechanistic hypothesis about the necessity of co-occurrence of efference copy and external stimulation. They assume that the external stimulus must arrive during the time of efference copy in order to have the attenuation effects. It implicitly assumes that the function of efference copy is online inhibition. Alternatively, reasoning from the quasi-perceptual experience of mental imagery, there should be a similar perceptual representation established during imagery. As long as the representation is in a similar format as perception, mental imagery should be able to create memory trace that influence on perception even after it ends. In fact, some mental

imagery studies have found adaptation effects in both visual (e.g. Ganis & Schendan, 2008 Neuroimage Visual mental imagery and perception produce opposite adaptation effects on early brain potentials; Wu et al., 2012 Frontiers The effects of visual imagery on face identification: an ERP study) and auditory domains (Scott, 2013 Psych Sci). If the authors did not observe the adaptation effect in sensorimotor domain, it could also be interesting as it suggesting the specific time constant of efferece copy for somatosensory and motor imagery. So again, for controlling imagery timing, attention and other confounds, as well as providing evidence for further functional aspects of efferece copy, I strongly recommend the authors run an adaptation version of their experiment by simply modifying the timing of their procedure.

We thank the reviewer for his/her suggestion. As stated in our response above, we have already controlled for divided attention in Experiment 2. To address his/her concerns about imagery timing and causality, we performed a new experiment with 12 new subjects in which we simply modified the timing of our procedure in line with the reviewer's suggestions.

In our new control experiment (Experiment 3), we included four conditions: three of them were identical to the conditions in Experiment 1: *base*, *press* and *imagine*. These conditions served to replicate the findings in Experiment 1. In the fourth condition (*imagine_{delay}*), we introduced a delay between the imagery task and the reference force. This fourth condition served to test whether participants would attenuate the reference force because of a memory trace from the imagery or if the imagery-driven attenuation requires the efferece copy to be generated at the same time as the reference force. We hypothesized that there would be no attenuation when the imagery task and the reference force did not overlap in time since this would break the causality between the imagined movement and its sensory consequences. It is important to note that previous studies have shown that this temporal requirement, which indeed suggests online inhibition, is necessary for the attenuation driven by overt movements^{1,2}.

Before the *imagine_{delay}* condition, we included a short session (10 min) to train the participants on pressing 2 N with their right index finger. Each trial of the *imagine_{delay}* condition then started with the participants imagining pressing 2 N for 3 seconds. After the 3 seconds of imagery, participants were verbally instructed to stop imagining the movement and remove their right index finger from the sensor. The reference force was then applied for 3 seconds. Finally, the participants were asked to reproduce the reference force with the slider. The interval between the end of the imagery phase and the application of the reference force was 5 seconds in order for the experimenter to give the verbal instruction and for the participants to stop the imagery task and remove their finger from the sensor. As in Experiments 1 and 2, we recorded the EMG from the right FDI to ensure that the participants were relaxed when imagining, and we administered the post-imagery questionnaire twice, once after each imagery condition.

The full procedures and detailed results from Experiment 3 are reported in **Supplementary Note 1** and **Supplementary Fig. 1**. The main finding is that we did

not observe any attenuation when the delay was introduced between the imagery task and the external stimulation (*imagine_{delay}* condition), suggesting that the two events need to overlap in time for attenuation to occur (*imagine_{delay}* versus *base*: $t(11) = -0.32$, $p = 0.755$, $CI = [-0.245, 0.183]$). Moreover, we replicated the effect of Experiment 1 with the same participants: the participants attenuated the reference forces when they imagined the movement at the same time the reference force was administered (*imagine* versus *base*: $t(11) = -2.67$, $p = 0.022$, $CI = [-0.362, -0.035]$).

We believe that the data from this new control experiment further strengthen our conclusions on the imagery-driven attenuation and eliminate any concerns about timing and causality.

References:

1. Blakemore, S. J., Frith, C. D. & Wolpert, D. M. Spatio-temporal prediction modulates the perception of self-produced stimuli. *J. Cogn. Neurosci.* **11**, 551–559 (1999).
2. Bays, P. M., Wolpert, D. M. & Flanagan, J. R. Perception of the consequences of self-action is temporally tuned and event driven. *Curr. Biol.* **15**, 1125–1128 (2005).

Reviewer #2 (Remarks to the Author):

1. I would suggest the authors to additionally include at least the following lines to the manuscript, which they they provided in response to one of my comments.

"We consider that our deduction that 'since both tasks (press and imagine) produce attenuation, then a common mechanism underlies both tasks' is very reasonable. From a theoretical perspective, we can see no alternative hypothesis for how attenuation would be produced in the imagery conditions without forward models and efference copies. In addition, a common underlying mechanism would be computationally less expensive than different mechanisms for covert and overt movements."

We thank the reviewer for his/her positive comments. As requested, we have now added text to our revised manuscript (Page 9, Lines 350-352).

REVIEWERS' COMMENTS:

Reviewer #1 (Remarks to the Author):

I appreciate the authors' effort on running the third experiment. Now I believe this has significantly strengthened this study. I only have some minor comments and suggestion.

1. Since the results in the new experiment are negative results, the authors need to use some Bayesian factor analyses to provide support for the null hypothesis.
2. A recent study in auditory domain found that the internal forward model in production can extend the prediction to basic sensory level, such as loudness perception [Tian, X., Ding, N., Teng, X., Bai, F., & Poeppel, D. (2018). Imagined speech influences perceived loudness of sound. *Nature Human Behaviour*, 1. doi:10.1038/s41562-018-0305-8]. This is very similar to the current study that indicate the internal linkage of motor and sensory systems using mental imagery paradigms. Citing this relevant study can make the current study and its argument much stronger.

Reviewer #2 (Remarks to the Author):

No further comments.

Reviewers' comments:

Reviewer #1 (Remarks to the Author):

I appreciate the authors' effort on running the third experiment. Now I believe this has significantly strengthened this study. I only have some minor comments and suggestion.

1. Since the results in the new experiment are negative results, the authors need to use some Bayesian factor analyses to provide support for the null hypothesis.

We thank the reviewer for his/her positive comments.

We have now calculated the Bayes factor for the comparison between the *base* and the *imagine_{delay}* conditions. Accordingly, the Bayes factor suggested that the observed data were 3.33 times more likely to have occurred under the null hypothesis than under the alternative hypothesis ($BF_{10} = 0.30 \pm 0.02\%$, default Cauchy prior width $r = 0.707$). This additional analysis further strengthens our conclusion on the critical importance of timing in sensory attenuation. We have added this additional analysis in the supplementary material.

2. A recent study in auditory domain found that the internal forward model in production can extend the prediction to basic sensory level, such as loudness perception [Tian, X., Ding, N., Teng, X., Bai, F., & Poeppel, D. (2018). Imagined speech influences perceived loudness of sound. Nature Human Behaviour, 1. doi:10.1038/s41562-018-0305-8]. This is very similar to the current study that indicate the internal linkage of motor and sensory systems using mental imagery paradigms. Citing this relevant study can make the current study and its argument much stronger.

We thank the reviewer for his/her suggestion.

In accordance with our responses during the first and second round of revisions, we consider that the sensory attenuation (described in our paper and mainly based on motor mechanisms) and the imagery aftereffects (described in the paper suggested by the reviewer and mainly based on stimuli repetition) are different phenomena. This is also reflected in the authors' viewpoint of the aforementioned paper; the authors do not refer to mechanisms of internal forward models or efference copies anywhere in their text. This is the reason why we opted not to include this particular paper in our large reference list. Nevertheless, we do cite several works in the auditory domain (Tian and Poeppel, 2010; Scott, 2013; Tian et al., 2016; Whitford et al., 2017) that are related to motor prediction and efference copy mechanisms and are therefore more relevant in our opinion.